# miR-200/375 control epithelial plasticity-associated alternative splicing by repressing the RNA-binding protein Quaking

Katherine A Pillman[1,†] , Caroline A Phillips[1,†], Suraya Roslan[1,†], John Toubia[1,†], B Kate Dredge[1], Andrew G Bert[1], Rachael Lumb[1], Daniel P Neumann[1], Xiaochun Li[1], Simon J Conn[1,2] , Dawei Liu[1], Cameron P Bracken[1,3], David M Lawrence[1], Nataly Stylianou[4], Andreas W Schreiber[1], Wayne D Tilley[5,6] , Brett G Hollier[4], Yeesim Khew-Goodall[1,3,7] , Luke A Selth[5,6] , Gregory J Goodall[1,3,7,*] & Philip A Gregory[1,3,**]

## Abstract

Members of the miR-200 family are critical gatekeepers of the epithelial state, restraining expression of pro-mesenchymal genes that drive epithelial–mesenchymal transition (EMT) and contribute to metastatic cancer progression. Here, we show that miR-200c and another epithelial-enriched miRNA, miR-375, exert widespread control of alternative splicing in cancer cells by suppressing the RNA-binding protein Quaking (QKI). During EMT, QKI-5 directly binds to and regulates hundreds of alternative splicing targets and exerts pleiotropic effects, such as increasing cell migration and invasion and restraining tumour growth, without appreciably affecting mRNA levels. QKI-5 is both necessary and sufficient to direct EMT-associated alternative splicing changes, and this splicing signature is broadly conserved across many epithelial-derived cancer types. Importantly, several actin cytoskeleton-associated genes are directly targeted by both QKI and miR-200c, revealing coordinated control of alternative splicing and mRNA abundance during EMT. These findings demonstrate the existence of a miR-200/miR-375/QKI axis that impacts cancer-associated epithelial cell plasticity through widespread control of alternative splicing.

**Keywords** alternative splicing; epithelial–mesenchymal transition; miR-200; miR-375; Quaking

**Subject Categories** Cancer; Cell Adhesion, Polarity & Cytoskeleton; RNA Biology

The EMBO Journal (2018) 37: e99016

## Introduction

The ability of cells to reversibly transition between epithelial and mesenchymal states (epithelial-to-mesenchymal transition, EMT) is exploited by tumours to drive malignant progression. EMT is governed by networks of transcriptional and post-transcriptional mechanisms. A reciprocal feedback loop between the miR-200 family and ZEB1/2 transcription factors plays a central role in epithelial cell plasticity (Bracken *et al*, 2008; Burk *et al*, 2008; Gregory *et al*, 2011), and by virtue of its potency functions widely in controlling cell invasiveness, stemness and tumour metastasis (Gibbons *et al*, 2009; Wellner *et al*, 2009; Brabletz & Brabletz, 2010). However, in addition to ZEB1/2, miR-200 can regulate many other target genes that contribute to these functions (Bracken *et al*, 2014, 2016; Perdigao-Henriques *et al*, 2016), although for most of these interactions the implications for cancer progression are not well understood.

Alternative splicing is an additional layer of post-transcriptional control that exhibits widespread changes during EMT and has causal effects on epithelial cell function (Warzecha *et al*, 2010; Shapiro *et al*, 2011). The epithelial spliced regulatory proteins (ESRP1 and ESRP2) play prominent roles in maintaining epithelial

1 Centre for Cancer Biology, University of South Australia and SA Pathology, Adelaide, SA, Australia
2 Flinders Centre for Innovation in Cancer, College of Medicine & Public Health, Flinders University, Adelaide, SA, Australia
3 Discipline of Medicine, The University of Adelaide, Adelaide, SA, Australia
4 Institute of Health and Biomedical Innovation, Australian Prostate Cancer Research Centre - Queensland, Princess Alexandra Hospital, Queensland University of Technology, Brisbane, Qld, Australia
5 Dame Roma Mitchell Cancer Research Laboratories, Adelaide Medical School, University of Adelaide, Adelaide, SA, Australia
6 Freemasons Foundation Centre for Men's Health, Adelaide Medical School, University of Adelaide, Adelaide, SA, Australia
7 School of Molecular and Biomedical Science, The University of Adelaide, Adelaide, SA, Australia
*Corresponding author. Tel: +61 8 8302 7751; E-mail: greg.goodall@unisa.edu.au
**Corresponding author. Tel: +61 8 8302 7829; E-mail: philip.gregory@unisa.edu.au
†These authors contributed equally to this work

alternative splicing patterns, and loss of ESRP expression in the mesenchymal state results in alterations to the these patterns (Warzecha *et al*, 2009; Brown *et al*, 2011). Several other RNA-binding proteins including RBFOX2, MBNL1/2, RBM47 and Quaking (QKI) have also been reported to directly influence mesenchymal associated alternative splicing (Shapiro *et al*, 2011; Venables *et al*, 2013a,b; Braeutigam *et al*, 2014; Yang *et al*, 2016); however, the direct contribution of these factors to alternative splicing and the mechanisms controlling their expression during EMT remain largely uncharacterised.

We find here that miR-200 exerts a widespread influence on alternative splicing during EMT, through its strong regulation of QKI. QKI is a member of the STAR family of RBPs and has been reported to have diverse functions in mRNA stability (Larocque *et al*, 2005; Zhao *et al*, 2006) and translation (Saccomanno *et al*, 1999; de Bruin *et al*, 2016b), miRNA processing (Wang *et al*, 2013, 2017) and alternative splicing (Hall *et al*, 2013; van der Veer *et al*, 2013; Zong *et al*, 2014; de Bruin *et al*, 2016a; Darbelli *et al*, 2017; Fagg *et al*, 2017; Hayakawa-Yano *et al*, 2017). We have recently shown that QKI can promote circular RNA formation during EMT (Conn *et al*, 2015), although the functional consequences of QKI in EMT remain unclear. While seeking to identify the miR-200 targets most biologically relevant to cancer progression, we surprisingly found that QKI is one of the most consistent clinical correlates of miR-200 activity. QKI is strongly regulated by miR-200 and miR-375, directly binds to and mediates hundreds of alternative splicing events, and regulates multiple facets of mesenchymal cell plasticity without significantly perturbing gene expression. Our findings implicate alternative splicing as an important regulator of cell plasticity and broaden the network of post-transcriptional changes orchestrated by miR-200. Given the important roles of miR-200, miR-375 and QKI in cell differentiation, we propose this pathway may define transcript selection in a broad range of biological contexts.

# Results

## QKI is inversely correlated with miR-200c in cancer

To identify miR-200c targets that may be especially relevant to cancer progression, we created a ranking method that merges microRNA–mRNA correlations from cancer and cell line data sets,

EMT data sets and Ago-HITS-CLIP data (Figs 1A and EV1A). The first and third highest ranked genes were ZEB1 and ZEB2, but surprisingly, the second highest ranked gene was the RNA-binding protein QKI (Fig 1B), which has not been described to be a target of miR-200, but was recently shown to regulate circRNA production in human mammary epithelial cells (HMLE) that have undergone EMT (Conn *et al*, 2015). To assess whether the negative correlation holds at the protein level, we measured QKI protein in a panel of breast cancer cell lines, which showed an even stronger negative correlation with miR-200c (Fig 1C).

Examination of QKI expression in breast cancer cohorts showed it was upregulated in basal-like and claudin-low subtypes, which display enhanced EMT-like features, and is indicative of poor distant metastasis-free survival (Figs 1D and EV1B and C). Moreover, in prostate cancer, *QKI* was elevated with increasing Gleason grade, in recurrent prostate cancers, and in metastases in several cohorts (Figs 1E and F, and EV1D). These data are consistent with QKI-mediating properties that promote tumour progression, such as cell migration and invasion, which are potently repressed by the miR-200 family (Bracken *et al*, 2014).

To assess whether the miR-200c–QKI inverse relationship is observed more broadly across other cancers, we examined the pan TCGA panel, which demonstrated QKI displays a strong negative recurrence score with miR-200c across the 10 represented tumour types (Fig 1G). Furthermore, when we examined all miRNAs for an inverse relationship with QKI across the 10 cancer types, we found the miRNAs with strongest negative recurrence with QKI are the members of the miR-200 family, along with miR-7 and miR-375 (Fig 1H). Together, these data demonstrate a consistent relationship between miR-200c and QKI in diverse experimental and clinical data sets, suggesting miR-200c may directly target QKI, with consequences for cancer progression.

## QKI-5 is directly targeted by miR-200c and miR-375

The QKI locus encodes three major isoforms, QKI-5, QKI-6, and QKI-7, named as such because the mRNAs were estimated to be 5, 6 and 7 kb, respectively (Ebersole *et al*, 1996). These isoforms differ in their carboxyl-termini and their 3′UTRs, but the 3′UTR polyadenylation sites were poorly characterised, with their RefSeq entries being discordant with the published sizes of the mRNAs. Consequently, to enable assessment of whether the negative

▶

**Figure 1. Quaking is a direct target and inversely correlates with miR-200c and miR-375 in cancer.**

A    Flow chart of pipeline used to discover miR-200c targets relevant to cancer.

B    Comparison of the top 20 miR-200c targets in listed experimental and clinical data sets ranked in order of consistency. An "X" in experimental data sets indicates evidence of miR-200 targeting. For clinical data sets (Liu *et al*, 2010; Taylor *et al*, 2010; Enerly *et al*, 2011), Pearson correlation coefficients between miR-200c and target genes are shown with significant values indicated in red.

C    Relative expression of QKI, E-cadherin and miRNAs in a human breast cancer panel of epithelial and mesenchymal cell lines.

D–F    Relative expression of QKI in tumour subtypes of the University of North Carolina 779 breast cancer set (Harrell *et al*, 2012), Gleason grades of the TCGA prostate cancers, and primary (P) versus metastases (M) from prostate cancer data sets listed by the first author (Lapointe *et al*, 2004; Varambally *et al*, 2005; Chandran *et al*, 2007; Tamura *et al*, 2007; Taylor *et al*, 2010; Grasso *et al*, 2012) shown as box-and-whisker plots. Box limits represent the 25th–75th percentiles with a median central line. For (D) and (E), whiskers extend to the minimum and maximum values with all data points shown. For (F), whiskers represent the 10th–90th percentiles. Expression differences were assessed by two-sample equal variance *t*-tests ***$P < 0.001$, **$P < 0.01$, *$P < 0.05$.

G, H    Recurrence scores of the top 20 target genes from (B), and the top 20 miRNAs that inversely correlate with QKI in the panTCGA cancer data set as calculated using CancerMiner (Jacobsen *et al*, 2013). Asterisks refer to minor form mature miRNA derived from the pre-miRNA (in this case miR-7-1-3p, miR-200b-5p and miR-200c-5p).

Source data are available online for this figure.

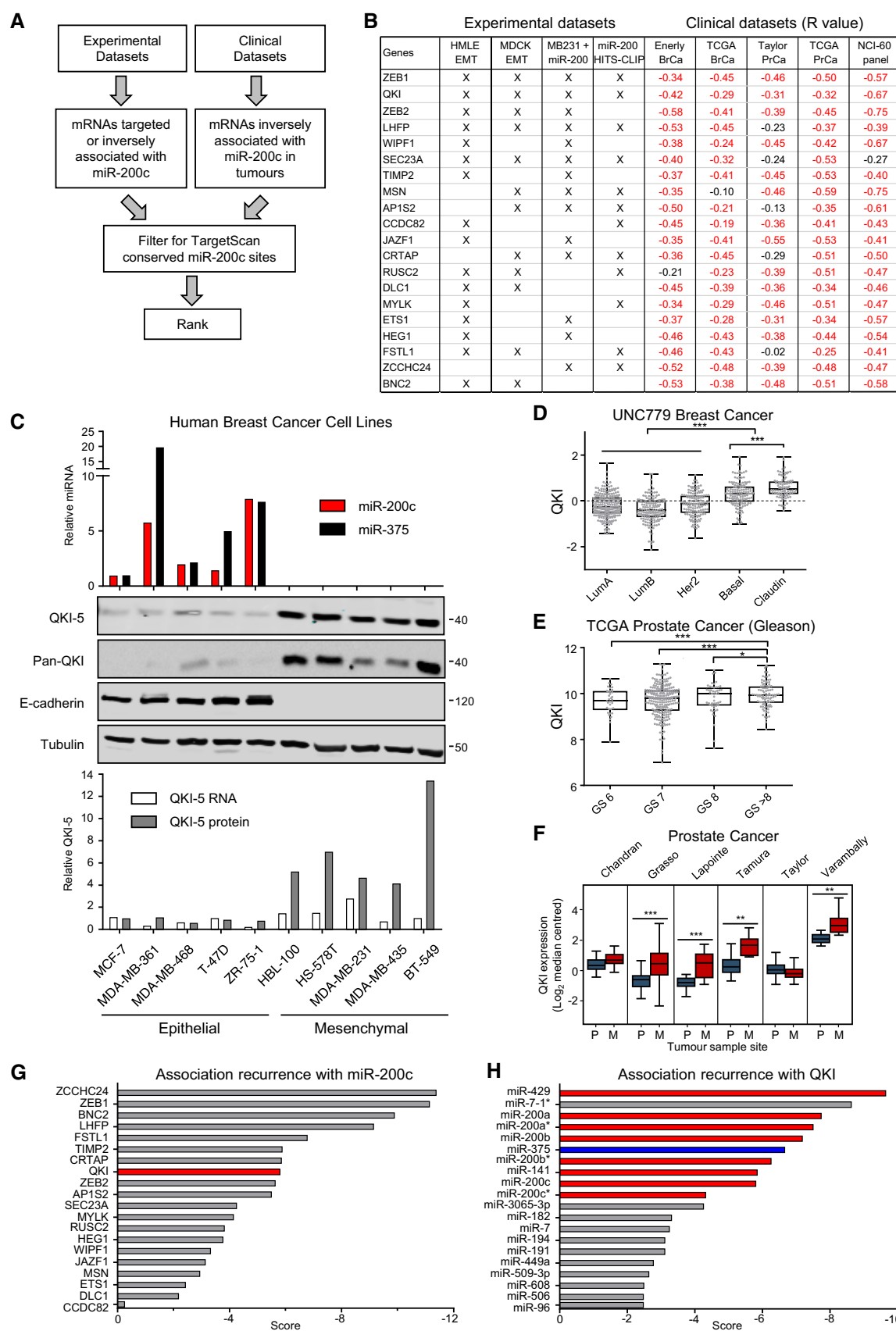

**Figure 1.**

correlation between miR-200c and QKI was due to direct regulation of QKI by miR-200c, we first characterised the 3′UTRs of each major isoform, using short- and long-read RNA-seq, and by RT–PCR using specific reverse transcription primers tiled across the annotated 3′UTRs (Fig EV2). This analysis showed that QKI-6 and QKI-7 have a similar 3′UTR, differentiated only by an additional segment at the beginning of the QKI-7 3′UTR, whereas QKI-5 has a completely distinct 3′UTR of 2.3 kb in length. Isoform-specific qRT–PCR showed that QKI-5 mRNA is expressed at a much higher level than QKI-6 and QKI-7 in breast cancer and TGF-β-treated mammary epithelial cells (Fig EV3A–D), and immunoblotting confirmed QKI-5 protein is much more abundant than the other isoforms (Fig 2A). The QKI-5 3′UTR has two predicted 8-mer binding sites for miR-200b/c that correspond to miR-200b peaks in our previous HITS-CLIP analysis of miR-200 binding sites (Bracken *et al*, 2014). In addition, we noticed the QKI-5 3′UTR also has multiple potential 7-mer sites for miR-375, which the association analysis had indicated is negatively correlated with QKI in multiple cancer types (Figs 1H and EV1E).

To verify that miR-200c and miR-375 regulate QKI levels, we examined the effect of overexpression of these miRNAs on QKI mRNA and protein, and on luciferase reporters bearing the QKI-5 3′UTR (Fig 2). Overexpression of miR-200c and miR-375 each reduced the level of QKI-5 and QKI-6 mRNAs and decreased all three QKI isoform proteins (Fig 2A). In contrast, miR-141, which differs from miR-200c by one nucleotide in its seed region, did not reduce QKI levels. Since QKI-5 is the major isoform in the breast cancer cells, we used the QKI-5 3′UTR to test for direct miRNA targeting. Both miR-200c and miR-375, but not miR-141, strongly repressed the full-length QKI-5 3′UTR reporter, replicating their effects on QKI expression (Fig 2B). Analysing the QKI-5 3′UTR proximal and distal regions in isolation revealed direct targeting of the distal region by the conserved miR-200c sites and the proximal region by two miR-375 sites, with repression of each abrogated by mutation of these sites (Fig 2B). These data demonstrate that miR-200c and miR-375 directly target the major QKI isoform by binding its 3′UTR and also directly or indirectly reduce the levels of the QKI-6 and QKI-7 isoforms.

### QKI-5 is dynamically regulated during EMT

Having confirmed QKI is regulated by miR-200 and miR-375, we assessed whether the expression of QKI is physiologically regulated during EMT, and in a reciprocal manner to these miRNAs. Consistent with its reciprocal relationship with miR-200 and miR-375 in cancer, QKI expression increased during TGF-β-induced EMT of human breast and canine kidney epithelial cells (Fig 3A) and during ZEB1-induced EMT of LNCaP human prostate cancer cells (Fig 3B). In each of these systems, QKI-5 protein was more responsive to the changes in miR-200c and miR-375 than was QKI-5 mRNA, consistent with the stronger effect on protein seen in Fig 2A. Examining this further, we observed that short-term inhibition of miR-200c or miR-375 increased QKI-5 protein in a dose-dependent manner with little effect on QKI-5 mRNA, suggesting these miRNAs especially affect translation of the QKI-5 mRNA (Fig 3C). Together, these findings show QKI-5 is robustly and dynamically regulated as cells transition between epithelial and mesenchymal states in a manner consistent with its control by miR-200 and miR-375.

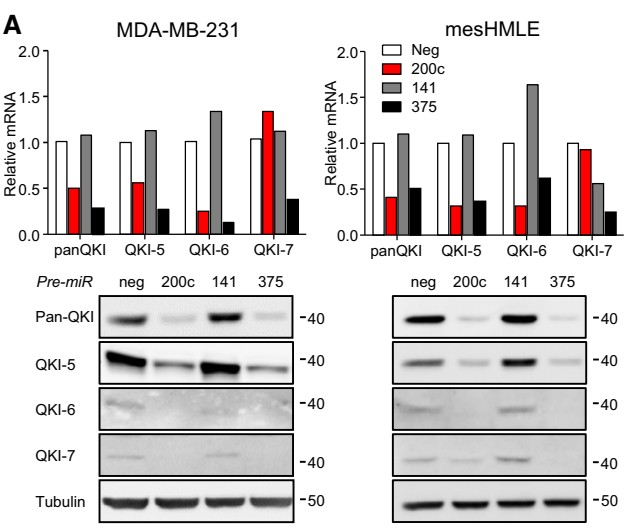

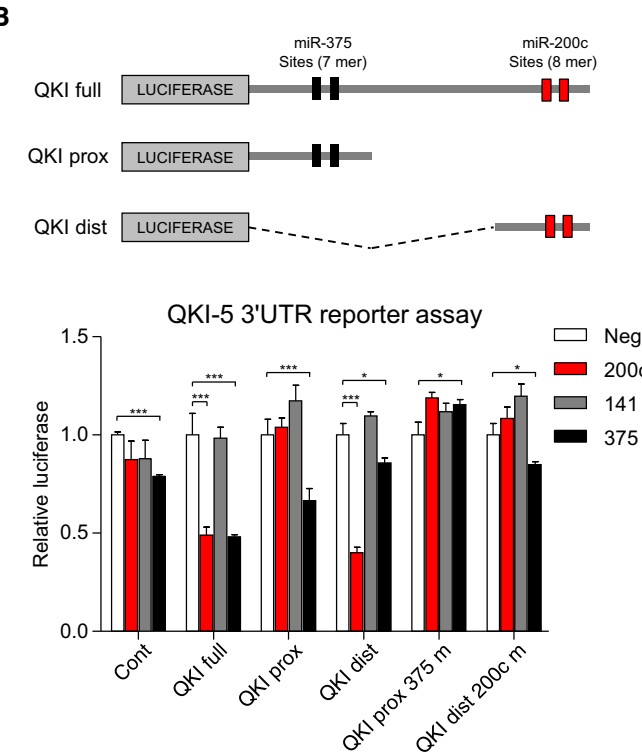

**Figure 2. Quaking is directly targeted and repressed by miR-200c and miR-375.**

A  Real-time PCR and Western blot of total QKI (panQKI) and individual QKI isoforms following transfection of MDA-MB-231 and TGF-β-treated mammary epithelial cells (mesHMLE) with miRNA precursors (Pre-miRs).

B  Schematic of the luciferase QKI-5 3′UTR reporter constructs indicating the miR-200c and miR-375 binding sites (top) and reporter assays of indicated QKI-5 3′UTR constructs with co-transfection of Pre-miRs into MDA-MB-231 cells (bottom). Constructs with mutations in the miR-375 or miR-200c sites are suffixed with "m". Data are represented as mean ± SD (n = 3). Significance was measured by two-tailed unpaired t-tests. *P < 0.05 and ***P < 0.001.

Source data are available online for this figure.

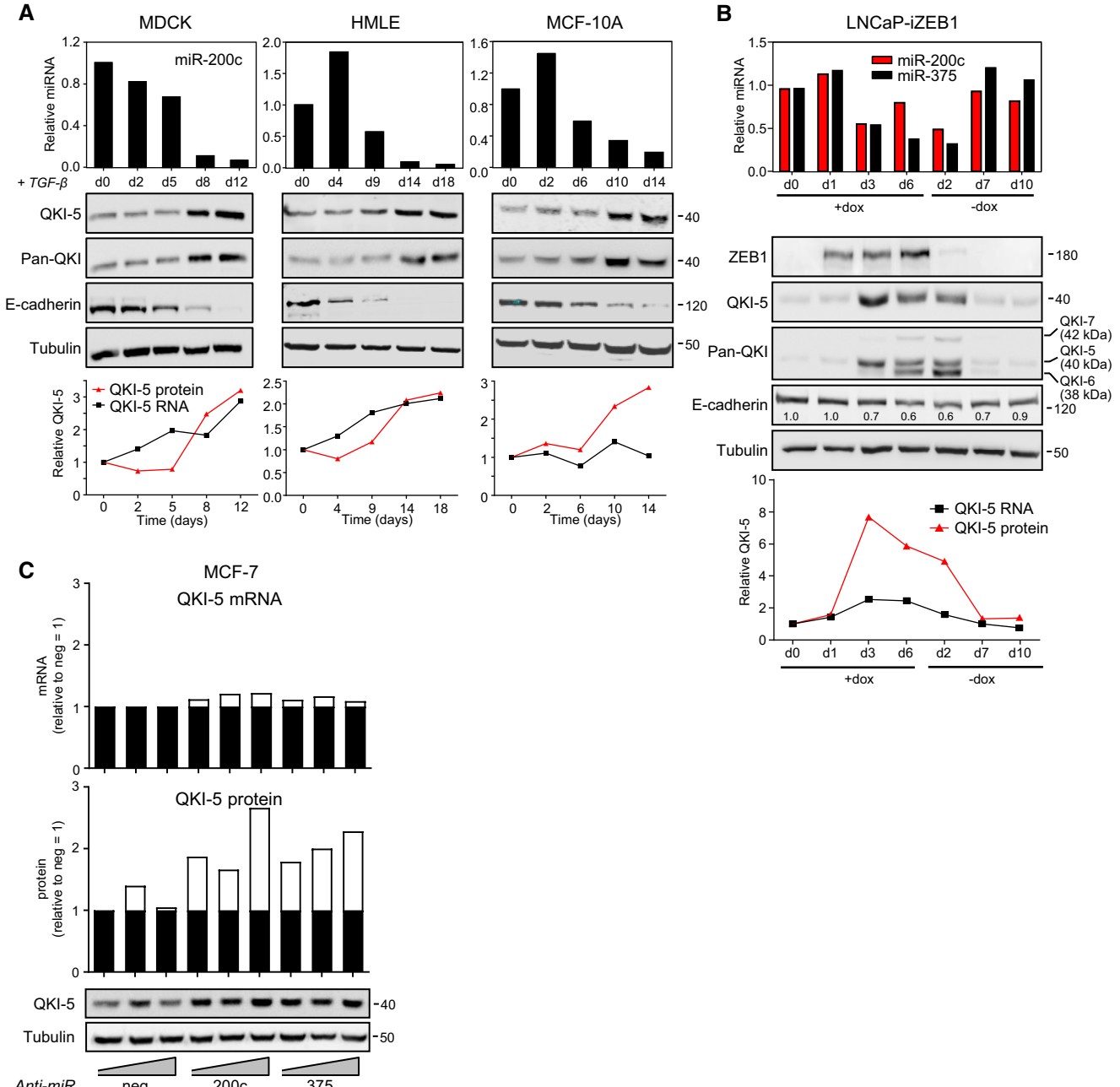

**Figure 3. Quaking is dynamically regulated during EMT/MET and is translationally repressed by miR-200c and miR-375.**

A   Epithelial cell lines were treated with TGF-β for the indicated time periods. Changes in expression of proteins were measured by Western blot, miR-200c by qPCR, with relative changes in QKI-5 mRNA and protein represented in graphs below.

B   LNCaP cells with inducible ZEB1 were treated with doxycycline for 6 days followed by its withdrawal for 10d. Changes in expression of proteins were measured by Western blot, miR-200c and miR-375 by qPCR, with relative changes in QKI-5 mRNA and protein represented in graphs below.

C   MCF-7 cells were transfected with increasing doses (20, 50 or 100 nM) of control (neg), miR-200c or miR-375 LNA inhibitors (anti-miR) for 3 days. Western blot of QKI-5 protein levels is shown and quantitated, along with QKI-5 mRNA by qPCR. White bars represent change in expression relative to the negative control at 20 nM.

Source data are available online for this figure.

## QKI-5 regulates epithelial cell plasticity and exerts pleiotropic effects on cell migration, invasion and tumour growth

Several studies have indicated QKI plays tumour suppressive roles in cancer (Chen *et al*, 2012; Zong *et al*, 2014; Bandopadhayay *et al*,

2016), but its ability to regulate EMT remains unclear. To establish whether the regulation of QKI-5 during EMT has consequences for EMT-related cell properties, we examined the effect of modulating QKI-5 on cell shape, and on migratory and invasive ability. Knockdown of QKI-5 caused MDA-MB-231 cells to change from a

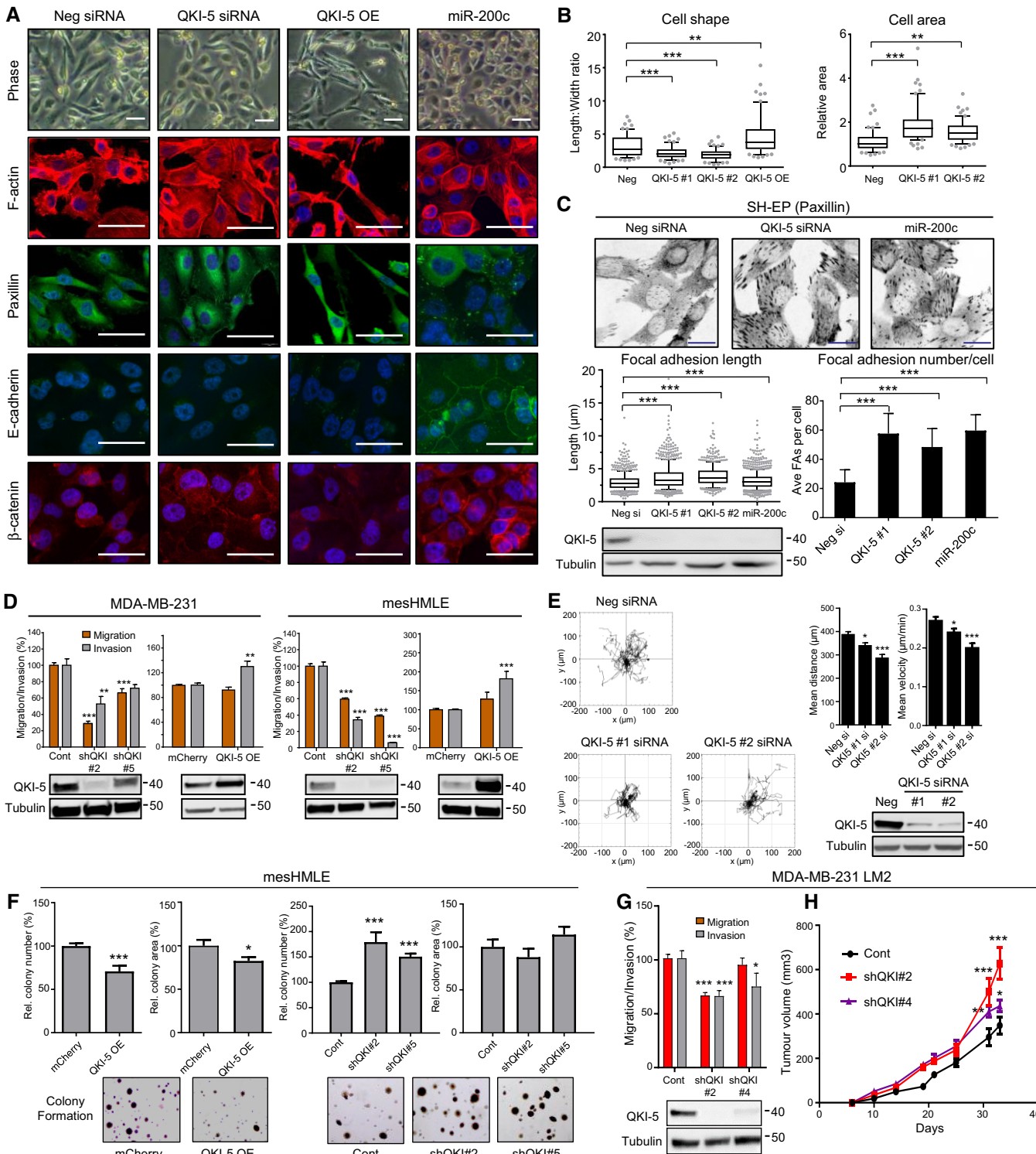

**Figure 4.**

spindle-shaped to a cuboidal morphology, with reduced length-to-width ratio and increased cell area, while QKI-5 overexpression had the opposite effect, promoting an even more spindle-shaped morphology with increased length-to-width ratio (Fig 4A and B). Moreover, reduction of QKI-5 phenocopied overexpression of miR-

200c in increasing focal adhesions, most clearly demonstrated in SH-EP neuroblastoma cells where changes in number and length were readily quantifiable (Fig 4A and C). However, QKI-5 knockdown was not sufficient to drive a full return to epithelial form, as evidenced by the lack of change in levels of multiple epithelial and

**Figure 4.   QKI-5 regulates epithelial cell plasticity and associated phenotypic properties.**

A   Phase contrast and fluorescent microscopy of F-actin, paxillin, E-cadherin and β-catenin in MDA-MB-231 cells transfected with control (neg), QKI-5 siRNA or miR-200c for 3 days. QKI-5-overexpressing (OE) MDA-MB-231 cells are also included in this panel. Cells are counterstained with DAPI (blue). Scale bars represent 50 μm.

B   Cell shape specified by length-to-width ratio and cell area were measured in 60 F-actin-stained cells from (A). Data are displayed as box-and-whisker plots. Box limits represent the 25th–75th percentiles with a median central line. Whiskers represent the 10th–90th percentiles. Points below and above the whiskers are shown. Significance between indicated groups was measured by a two-tailed Mann–Whitney test.

C   Paxillin staining of SH-EP cells transfected with control (neg), QKI-5 siRNA or miR-200c for 3 days. Focal adhesion lengths (min 500 counted) were calculated using at least 20 cells per treatment. Exact cell numbers and focal adhesion measured for each group are as follows: neg si (35, 698), QKI-5 #1 si (23, 766), QKI-5 #2 (24, 553) and miR-200c (31, 1,067). For focal adhesion length, data are displayed as a box-and-whisker plot and significance between indicated groups was measured by a two-tailed Mann–Whitney test. Box limits represent the 25th to 75th percentiles with a median central line. Whiskers represent the 10th–90th percentiles. Points below and above the whiskers are shown. Focal adhesion numbers were counted in individual discrete cells and graphed as focal adhesion numbers per cell (mean ± SD). Exact cell numbers for each group are as follows: neg si (22), QKI-5 #1 si (23), QKI-5 #2 (21) and miR-200c (24). Western blot with changes in QKI-5 levels is shown below.

D   Migration and invasion assays following shRNA-mediated knockdown (kd) or cDNA overexpression (OE) of QKI-5. Western blot with changes in QKI-5 levels is shown below. mCherry-expressing cell lines are shown as a control. Experiments were performed using at least three biological replicates and shown as mean ± SEM. Exact numbers for each group for both migration and invasion are as follows: MDA-MB-231 Control (10), shQKI#2 (7), shQKI#5 (3), mCherry (18) and QKI-5 OE (18); and mesHMLE Control (3), shQKI#2 (3), shQKI#5 (3), mCherry (6) and QKI-5 OE (3).

E   Non-directional migration of MDA-MB-231 cells following QKI-5-specific siRNA kd. Cell tracking of 25 cells per group was measured for 24 h and is depicted in windrose plots. Relative mean distance and velocity are shown as mean ± SD. Western blot with changes in QKI-5 levels is shown below.

F   Soft agar colony formation of mesHMLE cells with QKI kd or OE after 14 days (n = 9, mean ± SEM). Representative pictures of colonies are shown.

G   Migration and invasion assays of MDA-MB-231-LM2 cells following QKI kd (n = 3, mean ± SEM).

H   Growth of mammary tumours from MDA-MB-231-LM2 cells with QKI kd. Exact mouse numbers for each group are as follows: Cont (10), shQKI#2 (10) and shQKI#5 (9, one tumour did not grow). Data are graphed as mean ± SEM, and significance was calculated by two-way ANOVA with Dunnett's multiple comparisons test performed at each time point.

Data information: Unless otherwise indicated, significance was measured by two-tailed unpaired t-tests. P-values for all tests are indicated as *$P < 0.05$, **$P < 0.01$ and ***$P < 0.001$.

Source data are available online for this figure.

mesenchymal markers (Fig EV3E) and the failure to form E-cadherin and β-catenin-containing cell–cell junctions, which is in contrast to the effect of miR-200 (Fig 4A).

To examine the effect of QKI-5 on cell migration and invasion, we knocked down and overexpressed QKI-5 in MDA-MB-231 and mesHMLE cells and monitored the effect by transwell assay and by single-cell tracking. Mesenchymal cell migration and invasion were strongly reduced by QKI-5 knockdown, roughly in proportion to the effectiveness of the knockdown, while cell invasiveness was increased by QKI-5 overexpression (Fig 4D and E). SH-EP cell invasion was also reduced by QKI-5 knockdown (Fig EV3F), consistent with the stabilising effect seen on focal adhesions. Together, these data demonstrate QKI-5 stimulates changes in cell migration, invasion, focal adhesions and morphology, but unlike miR-200c, knockdown of QKI-5 does so without reverting cells to a fully epithelial phenotype.

As QKI has previously been implicated to have tumour suppressive roles, we tested its ability to regulate colony formation, a measure of anchorage independent cell growth. We found that mesHMLE cells robustly formed colonies in soft agar and that QKI knockdown enhanced, while QKI overexpression inhibited, this ability (Fig 4F). Therefore, in addition to enhancing cell migration and invasion (Fig 4D and E), QKI appears to concurrently reduce tumour formation capacity in mesHMLE cells. We further tested this in the tumorigenic MDA-MB-231 LM2 subline which also showed that while QKI-5 kd decreases cell migration and invasion, it significantly increased tumour growth (Fig 4G and H). These data indicate that QKI can exert multiple influences in cancer, potentially through regulating different target genes.

## QKI-5 has little effect on gene transcription during EMT, but has broad effects on alternative splicing

QKI plays diverse functions in RNA metabolism and therefore may conceivably regulate epithelial cell plasticity through several mechanisms. To assess the relevance of QKI-5 as a miR-200 target during EMT, we performed transcriptomic analysis of cells before and after initiation of EMT with TGF-β, and also assessed the transcriptomes of the resulting mesenchymal cells in response to miR-200c overexpression or to QKI-5 knockdown. RNA sequencing was performed on multiple replicates and at high depth to provide reliable quantitation of expression and alternative splicing (Table EV1). The expression of many genes was changed during the EMT, and many of these changes were reversed by restoration of miR-200c; however, most of the expression changes were not restored by knockdown of QKI-5 (Fig 5A). TGF-β and miR-200c had largely opposing effects on gene expression, significantly altering 2,677 and 2,656 genes, respectively (fold change > 2, q-value < 0.05), including archetypal epithelial (e.g. E-cadherin, occludin and desmoplakin) and mesenchymal (e.g. fibronectin, ZEB1 and ZEB2) genes (Figs 5A and C, and EV4A and Table EV2). In contrast, QKI-5 knockdown altered a much smaller subset of genes (787), and mostly to a lesser extent than either TGF-β or miR-200c, and did not cause significant changes in the hallmark EMT genes (Figs 5A and C, and EV4A and Table EV2). Furthermore, there was little consistency in the genes that were affected by QKI-5 knockdown in mesHMLE cells compared to knockdown in MDA-MB-231 cells (Fig EV4B and Table EV3). This was in contrast to miR-200c overexpression, which elicited similar changes in gene expression in both cell lines (Fig EV4C). Thus, the majority of transcriptomic alterations elicited by loss of miR-200 during EMT are not a result of QKI-5 upregulation.

In contrast, alterations to the relative levels of alternative splice isoforms that occurred during EMT were frequently counteracted by both overexpression of miR-200c and knockdown of QKI (Fig 5B). These commonly regulated alternative splicing events were predominantly comprised of exon skipping, exon inclusion and/or mutually exclusive exon usage (Fig 5D and Table EV4). Indeed, of the 1,138 alterations to skipped or included exon events that occurred in response to TGF-β-mediated EMT, 356 (or 31%) were regulated by

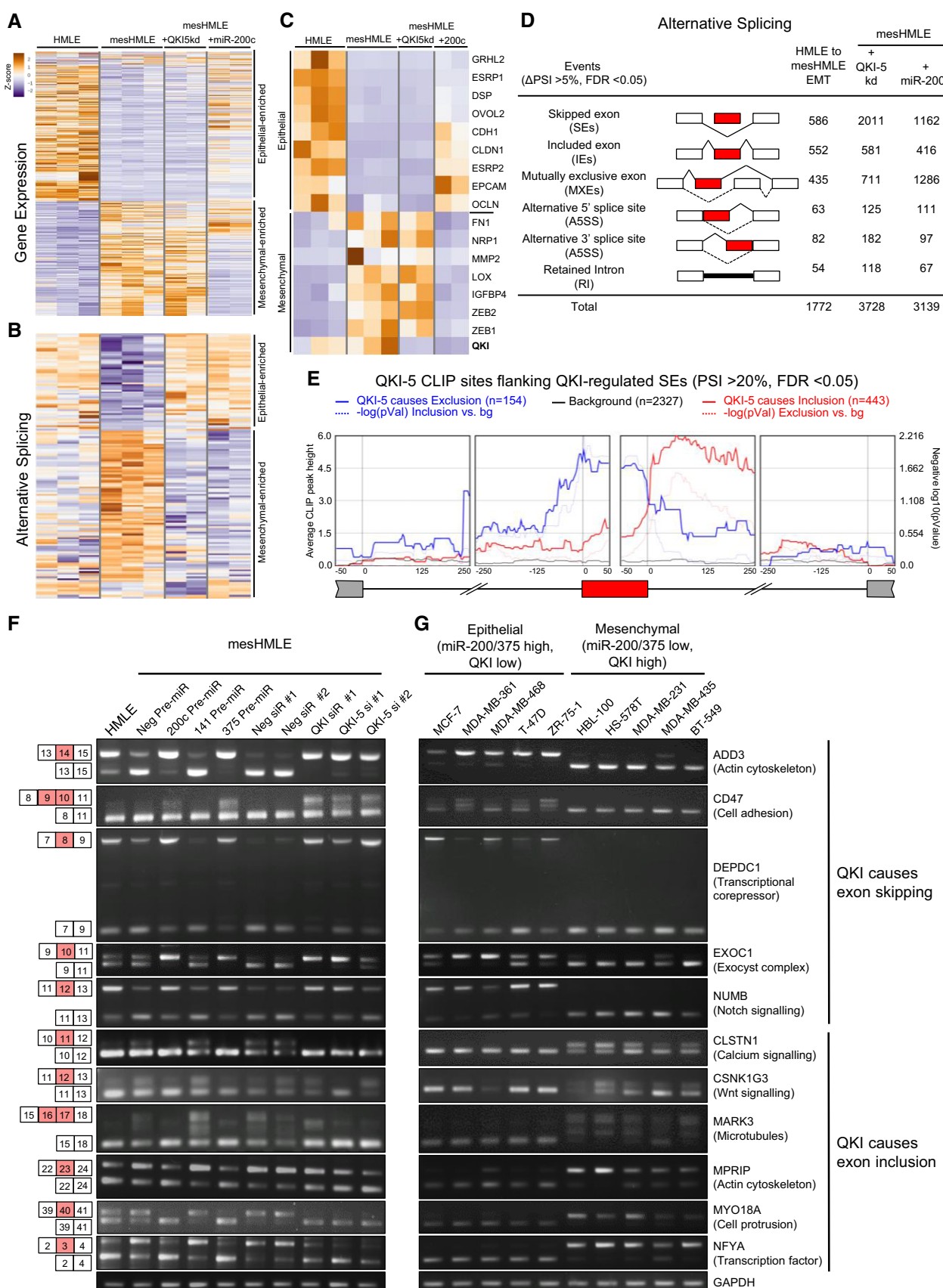

**Figure 5.**

**Figure 5.   QKI-5 directly binds to and controls alternative splicing of hundreds of EMT-associated splice events.**

A    Heat map depicting differential expression of individual samples from the HMLE, mesHMLE, mesHMLE + QKI-5 kd and mesHMLE + miR-200c groups. Genes that were detectably expressed (FPKM > 1) and whose change was at least twofold and statistically significant during EMT were included.

B    Heat map depicting relative changes in skipped and included exon events of individual samples from the HMLE, mesHMLE, mesHMLE + QKI-5 kd and mesHMLE + miR-200c groups. Events that are detected in all samples were utilised.

C    Heat map depicting differential expression of hallmark epithelial and mesenchymal genes in the HMLE, mesHMLE, mesHMLE + QKI-5 kd and mesHMLE + miR-200c groups.

D    Tabulation of alternatively spliced events altered during EMT (HMLE to mesHMLE) and with QKI-5 kd or miR-200c overexpression in mesHMLE cells.

E    QKI-CLIP map showing regional binding of QKI-5 in the flanking introns of skipped and included exon events regulated by QKI-5 knockdown with change in PSI > 20%, FDR < 0.05. The significance of enrichment versus background ($-\log10(P$-value)) is graphed as lighter shade lines. Numbers of events where QKI-5 knockdown causes inclusion or exclusion are indicated. The background measurement is calculated using all exons that are spliced, but not differentially spliced, in both mesHMLE and mesHMLE + QKI-5 kd.

F, G    PCR of QKI-regulated splice events during EMT and (F) in mesHMLE cells transfected with miRNAs or QKI siRNAs, and (G) in epithelial (QKI-low) versus mesenchymal (QKI-high) breast cancer cell lines. Exon annotations are denoted by the largest transcript. Major functions of the respective genes are indicated.

Source data are available online for this figure.

QKI-5 (Fig EV4D and Table EV5). These data indicate that the miR-200/QKI-5 axis is a major regulator of alternative splicing during EMT.

## Modulation of QKI-5 level is necessary and sufficient to alter splicing

To identify alternative splicing events in EMT that are directly regulated by QKI-5, we performed HITS-CLIP analysis of nuclear QKI-5 in mesHMLE cells. This technique identified a large number of potential QKI-5 binding sites across the transcriptome, predominantly within introns (Fig EV4E). Highlighting the robustness of the data set, motif analysis revealed strong enrichment of the QKI target motif, ACUAAC, and this sequence generally occurred near the centre of the HITS-CLIP peak (Fig EV4F and G). To explore how QKI-5 may function to promote skipping of some exons but inclusion of others, we plotted the average QKI-5 HITS-CLIP peak heights in the introns flanking the 597 exons whose exclusion or inclusion was strongly regulated by QKI-5 (change in percentage spliced in (ΔPSI) > 20%). This showed that exon skipping events were typically accompanied by QKI binding either within the skipped exon or within 100 nt upstream of the skipped 3′ splice site, whereas QKI-mediated exon inclusion involved binding of QKI within the downstream intron (Fig 5E).

To validate these findings, we selected eleven splicing events that were strongly regulated by QKI-5 and by miR-200c in the RNA-seq data and were also predicted to involve direct binding by QKI-5 (Fig EV5A). For each of these candidates, we used RT–PCR to monitor their splicing during EMT, and in mesHMLE and MDA-MB-231 cells treated with miRNAs or siRNAs targeting QKI (Figs 5F and EV5B). We also examined the effects of miR-375, which targeted QKI-5 to a similar extent as miR-200c. Overexpression of miR-200c and miR-375 and knockdown of QKI each reversed the splicing profiles observed during EMT for all events tested (Figs 5F and EV5B). By contrast, overexpression of miR-141, which does not target QKI-5, had no effect on the splicing events. Furthermore, in a panel of breast cancer cell lines, the reciprocal levels of both miR-200 and miR-375 compared to QKI effectively demarcated the splicing profile of these events (Fig 5G). Collectively, these data provide strong evidence that a miR-200/miR-375/QKI axis broadly influences EMT-associated splicing patterns.

To assess the extent to which QKI is necessary or sufficient to regulate these splicing events, we treated HMLE cells with TGF-β

while simultaneously knocking down QKI-5 levels and examined specific EMT-associated splicing events by PCR. Knockdown of QKI robustly blocked the dynamic splicing changes that occured during EMT, but did not alter changes in the EMT marker genes E-cadherin and fibronectin, demonstrating that its upregulation was specifically required for EMT-induced alternative splicing (Figs 6A and EV5C). To determine whether miR-200c and miR-375 could induce splicing alterations in the absence of an ability to target QKI-5, we overexpressed a non-targetable form of QKI-5 (i.e. lacking its 3′UTR) in mesHMLE cells and in the presence or absence of ectopic delivery of these miRNAs. QKI-5 overexpression augmented the degree of mesenchymal splicing and completely abrogated the ability of miR-200c and miR-375 to induce epithelial splicing patterns (Fig 6B). As a further control, we showed that the QKI coding region siRNA, but not the QKI-5 3′UTR siRNA, repressed exogenous QKI-5 and reversed these splicing changes (Fig 5B). Together, these data demonstrate that QKI-5 is the primary target by which these miRNAs regulate EMT-associated alternative splicing events.

We next assessed the generalisability of these findings in other EMT contexts. Firstly, we knocked down QKI in the LNCaP-iZEB1 model and found it was similarly essential for EMT-induced splicing events but did not influence the level of archetypal EMT markers (Fig 6C). To test whether QKI-5 induction is sufficient by itself to drive EMT-associated splicing changes, we generated doxycycline-inducible QKI-5-expressing prostate (LNCaP) and breast (MCF-7 and HMLE) cell lines. In all cases, QKI-5 induction was associated with dynamic changes in splicing of QKI-5 target genes, which were lost upon withdrawal of doxycycline, and these changes occurred in the absence of notable effects on cell morphology and EMT marker expression (Figs 6D and E, and EV5D). Remarkably, QKI-5-regulated splicing occurred without EMT inducers or the accompaniment of other EMT effectors, indicating that QKI-5 levels alone are a major determinant of splicing outcomes. Collectively, these data demonstrate that miR-200c and miR-375 limit mesenchymal splicing by repressing a single target, QKI-5, which is sufficient to direct an EMT-associated alternative splicing programme.

## *De novo* identification of QKI-regulated alternative splicing reveals its widespread activity in cancers

To assess the relevance of QKI-mediated alternative splicing during EMT to human cancers, we initially evaluated the relationship between *QKI* expression and specific splicing events in TCGA breast

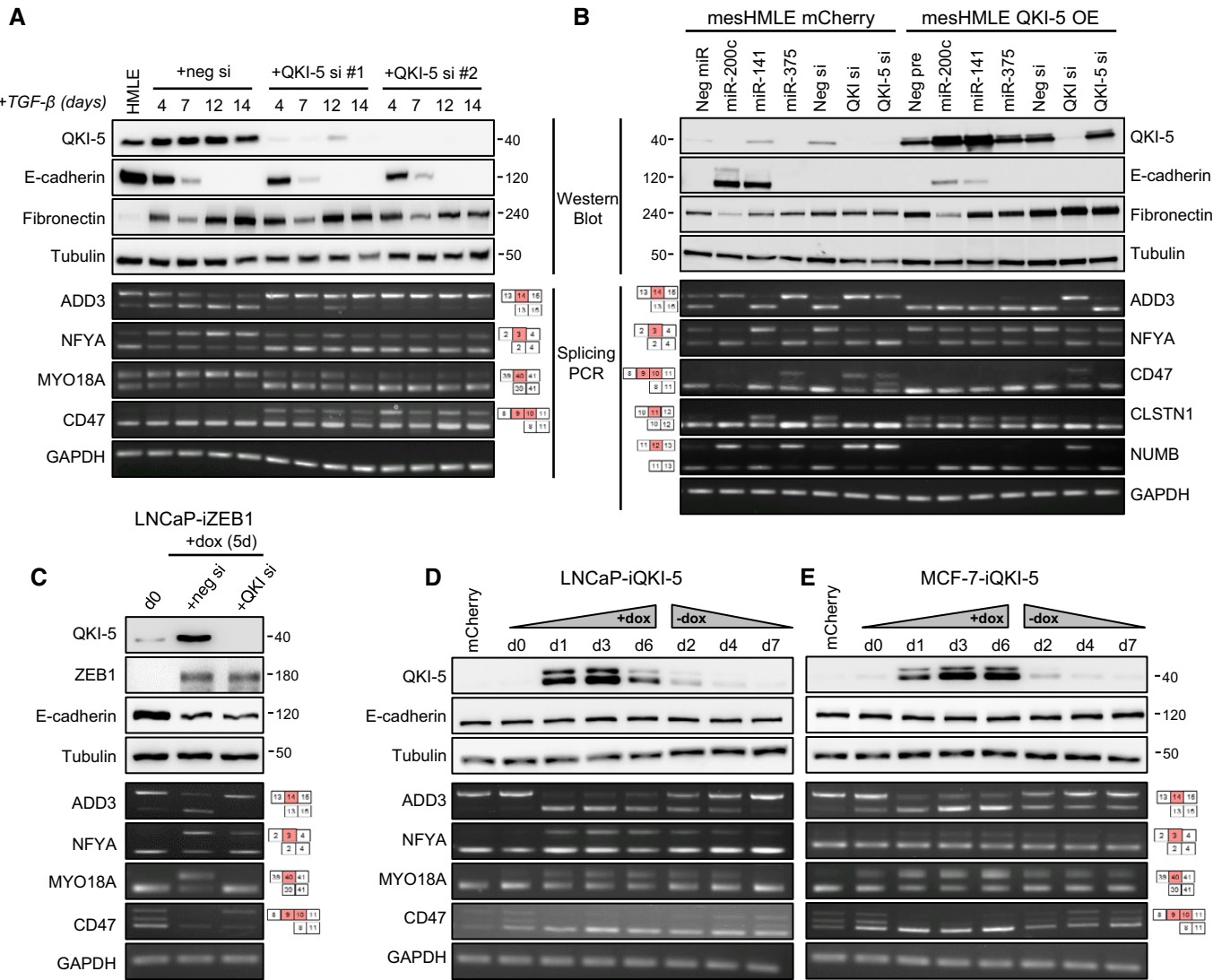

**Figure 6. QKI is sufficient and necessary for EMT-associated splicing.**

A–E    Western blot of EMT markers and PCR of QKI-regulated alternative splicing events are shown following (A) TGF-β induction of EMT in HMLE cells transfected with control or QKI-5 siRNAs, (B) transfection of mCherry- or QKI-5-overexpressing (OE) mesHMLE cells with miRNA or QKI siRNAs for 3 days, (C) doxycycline induction of ZEB1 in LNCaP-iZEB1 cells transfected with control or QKI siRNA for 5 days, and (D, E) doxycycline induction of QKI-5 and subsequent withdrawal in LNCaP- and MCF-7-iQKI-5 cells, with mCherry-expressing cell lines shown as an additional control.

Source data are available online for this figure.

cancer RNA-seq data. Supporting our *in vitro* findings, the splicing of all genes verified to be directly regulated by QKI was highly correlated with QKI levels (Fig 7A). We then constructed a metric to quantify the global influence of QKI on alternative splicing in breast cancer: More specifically, the PSI for every splice event in the transcriptome was calculated for the 10% of samples with the highest and lowest level of QKI. Plotting the difference in PSI (ΔPSI) versus the statistical significance of the difference revealed several hundred splice events that were highly dependent on the QKI level, including all 20 of the genes that we had identified as the 20 most regulated by QKI during EMT *in vitro* (Fig 7B and Table EV6). This demonstrates that the alternative splice events most strongly associated with QKI expression in breast cancers align closely with the

QKI-regulated events we had identified during EMT of cell lines, and more broadly provides strong evidence that QKI is a prominent driver of splicing changes in breast cancer.

To assess whether the influence of QKI on splicing in cancer extends to carcinomas more generally, we repeated the analysis on seven epithelial cancers for which there are at least 450 samples present in the TCGA. To visualise the results, we performed unsupervised clustering of the ΔPSI values (Fig 7C), which revealed a cluster of QKI-responsive alternative splicing events that are common across all cancer types, and a cluster of other QKI-responsive events that are cancer-type-specific. The genes for which we identified QKI binding sites proximal to the splice sites by QKI-HITS-CLIP were predominantly in the conserved cluster (Fig 7C),

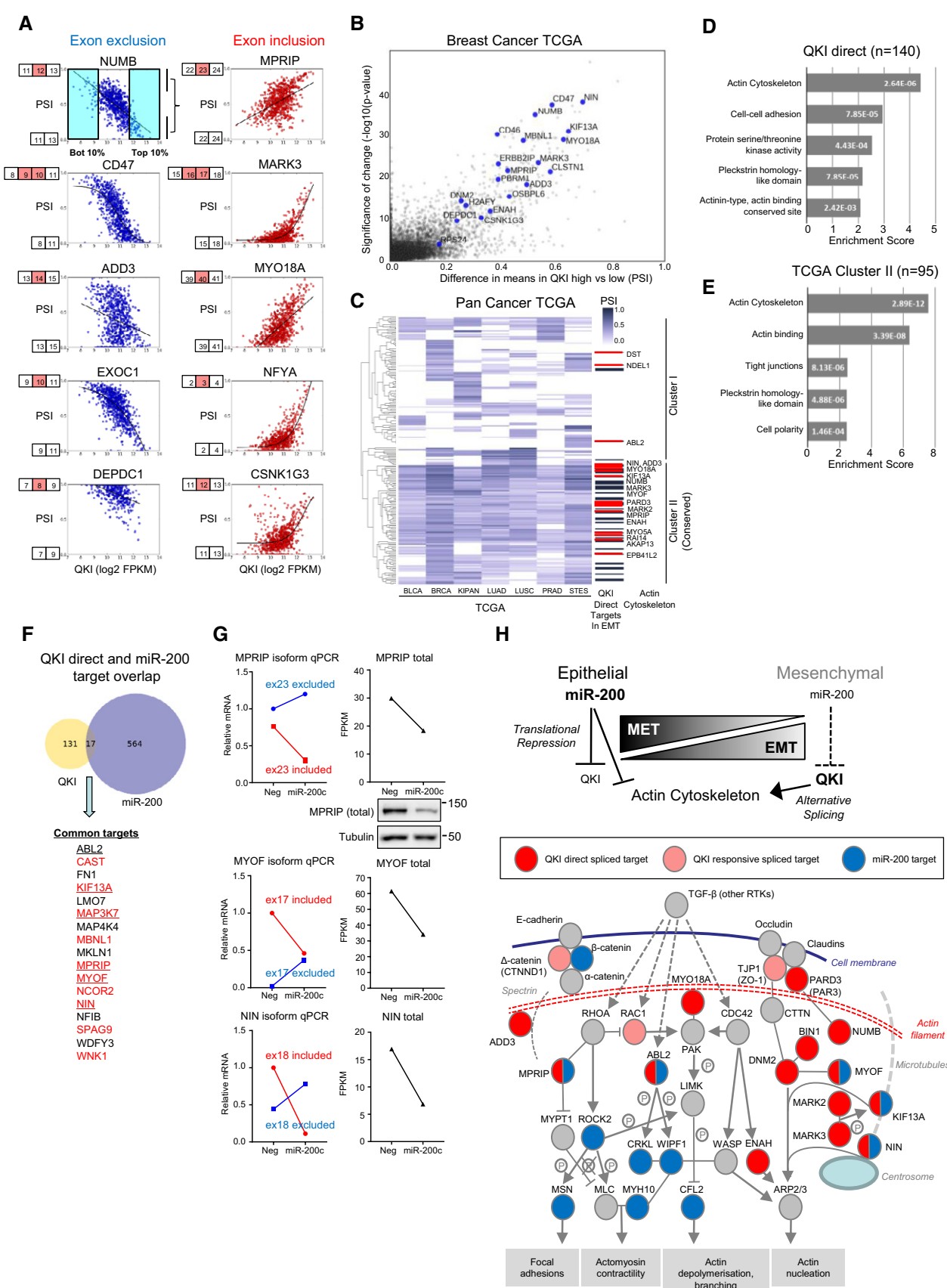

**Figure 7.**

**Figure 7.  miR-200–QKI coordinates splicing and expression changes in the actin cytoskeletal network in cancer.**

A    Representative graphs of splice events identified as QKI-responsive in breast TCGA data. The top left graph illustrates the methodology used for *de novo* identification of QKI-regulated events where mean PSI differences between the samples with the highest and lowest deciles of QKI expression were calculated.

B    Plot of strength of QKI-mediated splicing changes from TCGA breast cancer data versus statistical significance of the change. The most significant alternative splicing event for each gene is plotted. The top 20 genes identified as having splicing regulated by binding of QKI during EMT (as in Fig 5B) are labelled.

C    Heat map showing clustering of QKI-responsive alternatively spliced events across epithelial TCGA cancers. Genes identified as having splicing regulated by binding of QKI during EMT (as in Fig 5B) are marked with black bars or, where such genes have functions related to actin cytoskeleton dynamics, are marked with red bars and are named.

D    Gene ontology analysis of genes identified as having splicing regulated by binding of QKI during EMT.

E    Gene ontology analysis on genes exhibiting QKI-responsive alternative splicing changes in multiple cancers (cluster II in Fig 7C).

F    Venn diagram of overlap between genes that exhibit QKI-directed splicing changes during EMT and are directly bound by miR-200a/141 or miR-200b/200c (as identified by miR-200 Argonaute HITS-CLIP (Bracken *et al*, 2014)). The genes in the intersection set are named, with those having a function related to actin cytoskeleton dynamics underlined and those whose splicing is broadly conserved in carcinomas shown in red.

G    Quantitation of alterative splicing changes (qPCR) and total transcript levels (from RNA-seq, FPKM) following miR-200c transfection of mesHMLE cells for 3 days. Western blot of total MPRIP levels in these samples is also shown.

H    Actin cytoskeletal pathway adapted from Bracken *et al* (2014) with miR-200 direct targets highlighted in blue, QKI direct targets in red and QKI-responsive targets in pink. Major outputs on actin dynamics are indicated.

Source data are available online for this figure.

reinforcing the conclusion that QKI influences alternative splicing in all of these cancers.

## miR-200c coordinately regulates gene expression and alternative splicing of actin cytoskeleton-associated targets

The studies described above demonstrated that QKI affects the alternative splicing of numerous genes during EMT, for some genes causing a relatively small change in the balance of splice isoforms, but for some genes causing a more drastic change. We also found that QKI strongly affects the morphology of cells as well as their capacity to migrate and invade, which suggests that genes involved in these processes may be alternatively spliced in response to QKI. To assess this possibility, we performed gene ontology (GO) analysis on the set of genes whose change in splicing during EMT was reversed by knockdown of QKI (Appendix Fig S1). The most enriched functional group was genes involved in cell–cell adhesion followed by genes involved in serine/threonine kinase activity, both of which are consistent with changes that can affect cell motility. To apply a more stringent criterion to the gene list, we restricted it to only include genes that were confirmed to be direct QKI targets in the QKI-5 HITS-CLIP analysis. This filtered gene list retained cell–cell adhesion and serine/threonine kinase activity as highly enriched functional groups, but the most significantly enriched group was genes involved in regulation of the actin cytoskeleton (Fig 7D). We also performed GO analysis on the set of genes whose alternative splicing was consistently affected by QKI in different carcinomas (Fig 7C), which confirmed that regulation of the actin cytoskeleton was the function most affected by QKI (Fig 7E).

Previous work from our laboratory has demonstrated that miR-200 directly targets multiple genes involved in the actin cytoskeleton regulatory network (Bracken *et al*, 2014). The observation that genes alternatively spliced by QKI during EMT and in cancer are also frequently members of the actin cytoskeleton network identifies a novel convergence of miR-200 function, whereby it controls both alternative splicing of this network (indirectly, via QKI) and gene expression (directly, via 3′UTR targeting). To identify genes regulated by both mechanisms, we merged Ago-miR-200 HITS-CLIP targets (Bracken *et al*, 2014) with

QKI-5 HITS-CLIP targets, identifying 17 genes subject to both direct miR-200 and direct QKI-5 regulation (Fig 7F). Remarkably, 10 of these genes exhibit highly conserved QKI-responsive splicing changes across all cancer types (Fig 7F, shown in red). We tested three actin-associated genes (MPRIP, NIN and MYOF) from this list and found that miR-200c caused a dramatic switch in their isoforms while simultaneously reducing their overall expression (Fig 7G). These genes form parts of actin-associated sub-networks (Bracken *et al*, 2014), with many being direct miR-200 targets, QKI-5 targets or targets of both factors (Fig 7H). Collectively, these data demonstrate the existence of a miR-200–QKI-5 regulatory pathway that coordinates widespread changes in gene expression and alternative splicing of actin-associated gene networks in cancer.

## Discussion

It is well recognised that the miR-200 family are primary enforcers of the epithelial phenotype through restraining expression of the ZEB1/2 transcription factors (Burk *et al*, 2008; Gregory *et al*, 2008; Korpal *et al*, 2008; Park *et al*, 2008). ZEB1 in particular invokes transcriptional reprogramming that drives EMT, and associated influences on tumour cell invasion, stemness and metastatic potential (Wellner *et al*, 2009; Chaffer *et al*, 2013; Lehmann *et al*, 2016; Krebs *et al*, 2017). Despite the potency of this double-negative feedback loop leading to mutually exclusive expression of miR-200 and ZEB1 (Gregory *et al*, 2008; Brabletz & Brabletz, 2010), miR-200 targets many other transcripts with potential relevance to cancer progression (Bracken *et al*, 2014, 2016; Perdigao-Henriques *et al*, 2016). Here, we demonstrate that miR-200 constrains widespread EMT-associated alternative splicing changes through translational repression of a single RNA-binding protein QKI-5. QKI-5 is sufficient to mediate plasticity between epithelial and mesenchymal states by directing mesenchymal splicing alterations, especially within actin cytoskeleton-associated genes, in the absence of major changes to total mRNA levels. Thus, by targeting ZEB1 and QKI-5, miR-200 maintains epithelial cell identity by coordinately limiting transcriptional and alternative splicing programmes that would otherwise promote epithelial–mesenchymal plasticity.

We found QKI-5 is also suppressed by a second miRNA, miR-375, which is known to be enriched in certain epithelial cells and can maintain epithelial features (Ward *et al*, 2013; Selth *et al*, 2017), but also plays prominent roles in regulating pancreatic cell function and endocrine signalling (Poy *et al*, 2004, 2009). Interestingly, miR-375 is also subject to transcriptional repression by ZEB1 (Selth *et al*, 2017). This suggests that ZEB1 acts within a feedback loop to increase QKI-5 levels and enhance mesenchymal splicing by coordinately limiting both miR-375 and miR-200 expression. As both miR-200c and miR-375 act both on the mRNA level and on translation of QKI-5, this ZEB1/miR-200/miR-375/QKI-5 regulatory loop is particularly strong, producing the strongly mesenchymal expression pattern of QKI we observed in the panel of breast cancer cell lines that we examined, paralleling the mutually exclusive expression of miR-200c and ZEB1 (Gregory *et al*, 2008). Since QKI has notable functions outside of EMT, including regulating neuronal (Larocque *et al*, 2005; Zhao *et al*, 2006; Hayakawa-Yano *et al*, 2017), smooth muscle and vascular (Noveroske *et al*, 2002; Li *et al*, 2003; van der Veer *et al*, 2013; Cochrane *et al*, 2017) cell function, and alternative splicing programmes during muscle cell (Hall *et al*, 2013) and monocyte (de Bruin *et al*, 2016a) differentiation, we propose that the ZEB1/miR-200c/miR-375/QKI-5 pathway may influence splicing outcomes that impact a broad range of cell differentiation contexts.

Although several studies have indicated QKI is involved in cancer-associated splicing, its action to date has largely been inferred from motif analysis or from effects on single target genes, with the exception of a global analysis in lung cancer cells (Novikov *et al*, 2011; Zong *et al*, 2014; Danan-Gotthold *et al*, 2015; de Miguel *et al*, 2016; Yang *et al*, 2016). By combining global QKI-5 CLIP-seq with RNA-seq analysis in an EMT system, we identified 242 skipped exon events (in 140 genes) directly regulated by QKI-5 (Table EV7), with direct and indirect alternative splicing targets constituting approximately one-third of EMT alternative splicing events (Fig EV4D). Importantly, QKI-5 was both sufficient and necessary to drive EMT-associated alternative splicing, even in the absence of EMT inducers. The observation that where QKI promotes exon skipping, it is via binding at or near the skipped 3′ splice site is consistent with a model in which QKI competes with U2 snRNA for the branch point, since they have the same sequence determinants for binding.

Together, our data demonstrate that dynamic changes in QKI-5 expression that occur during EMT and MET are a major determinant of mesenchymal splicing programmes. In contrast to QKI-5, ESRP1 and ESRP2 are strongly downregulated during EMT and drive epithelial splicing programmes when re-introduced into mesenchymal cells (Warzecha *et al*, 2009; Brown *et al*, 2011; Shapiro *et al*, 2011). A recent study indicated ESRP1/2 and QKI could co-regulate and induce opposing alternative splicing in at least twelve splice events, including CLSTN1, MPRIP, EXOC1 and DEPDC1 which we describe here to be QKI-regulated (Yang *et al*, 2016). Furthermore, ESRP1 and QKI-5 were respectively the most up- and downregulated splicing factors in response to miR-200c expression in mesenchymal cells (Appendix Fig S2), likely owing to loss of ZEB1-mediated transcriptional repression of ESRP1/2 (Horiguchi *et al*, 2012; Preca *et al*, 2015) and direct targeting of QKI-5. Collectively, our observations indicate that the miR-200-ZEB loop coordinates an opposing balance of QKI-5 and ESRP1 expression to orchestrate widespread EMT-associated alternative splicing programmes. However, it is likely that other splicing factors also implicated in EMT, such as RBFOX2,

MBNL1/2 and RBM47, work cooperatively with QKI-5 and ESRP1 to fine-tune alternative splicing patterns (Shapiro *et al*, 2011; Venables *et al*, 2013a,b; Braeutigam *et al*, 2014; Yang *et al*, 2016; Neumann *et al*, 2018).

By comparing exon usage and QKI expression in tumours, we found that the QKI alternative splicing signature is conserved across a range of epithelial-derived tumours, indicative of their direct contributions to cancer progression. QKI-regulated alternative splicing events are enriched with functions involving control of actin cytoskeletal dynamics, with the splicing of ENAH (MENA) and NUMB directly linked to several EMT-associated properties (Di Modugno *et al*, 2012; Lu *et al*, 2015). Other genes such as CD44 and EXOC7 exhibit profound changes in splicing that are required for actin dynamics and/or EMT, but are regulated by ESRP1 rather than QKI-5 (Brown *et al*, 2011; Lu *et al*, 2013). Here, we show that modulation of QKI-5 levels exerts pleiotropic effects on mesenchymal cells including changes to cell morphology, migration, invasion, focal adhesions and tumour growth. Collectively, these QKI-5-regulated phenotypes are consistent with clinical data where QKI-5 is enriched in breast and prostate cancers with increased invasive and metastatic potential. Interestingly, several other studies have indicated that QKI can also play tumour suppressive roles (Chen *et al*, 2012; Zong *et al*, 2014; Bandopadhayay *et al*, 2016), and in one case, QKI-directed alternative splicing of NUMB has been shown to reduce growth of lung cancer cells *in vivo* (Zong *et al*, 2014). Thus, QKI may play a dual role in cancer progression, limiting the initial formation of tumours but increasing their subsequent invasiveness through regulating the splicing of different target genes, such as through promotion of mesenchymal ENAH splicing. It is also important to note that although QKI-5 knockdown produced modest and inconsistent effects on gene expression (Fig EV4), we cannot rule out that QKI-5 may also directly or indirectly influence specific mRNA or miRNAs to regulate different aspects of cancer progression, as has been reported in other contexts (Chen *et al*, 2012; Lu *et al*, 2014; Yu *et al*, 2014; Shingu *et al*, 2017; Zhou *et al*, 2017). Similarly, QKI-5 also regulates circRNA formation during HMLE cell EMT (Conn *et al*, 2015), and although the functions of these molecules remain poorly understood, they could contribute to epithelial–mesenchymal plasticity.

In addition to modulating alternative splicing through QKI-5, miR-200 directly represses expression of many genes involved in regulating actin cytoskeletal dynamics (Bracken *et al*, 2014). By intersection of direct miR-200 and QKI bound transcripts, we provide evidence that a subset of these genes are co-regulated by both miR-200c and QKI-5. This finding represents a novel mechanism of coordinate control of both the abundance and alternative splicing of specific factors, which to our knowledge has not been reported previously. Of the three genes we assessed, MPRIP and MYOF have been previously verified as miR-200 targets and directly influence cell migration and invasion (Li *et al*, 2012; Bracken *et al*, 2014; Volakis *et al*, 2014), but the expression and function of splice variants of these proteins have not been assessed. More broadly, miR-200 and QKI-5 regulate the expression and/or alternative splicing of a number of genes which directly interact with each other and integrate into signalling networks that drive cytoskeletal changes influencing cell morphology, migration and invasion (Fig 7H). Several of these alternatively spliced products including ENAH (MENA), PARD3 and NIN display altered phosphorylation,

protein–protein interactions and cell localisation, respectively (Gao *et al*, 2002; Balsamo *et al*, 2016; Zhang *et al*, 2016). A future challenge will be assessment of the functions of these isoforms and their impact on epithelial plasticity.

# Materials and Methods

### Cell culture, transfection and production of stable cell lines

Human breast cancer cell lines were cultured as described previously (Gregory *et al*, 2008). HMLE cells (Mani *et al*, 2008) were cultured in HuMEC Ready Media (ThermoFisher) and induced to undergo EMT by transferring to DMEM:F12 media (1:1) supplemented with 10 μg/ml insulin, 20 ng/ml EGF, 0.5 μg/ml hydrocortisone and 5% foetal calf serum (FCS) and treating with 2.5 ng/ml of TGF-β1 (R&D) for at least 14 days. MesHMLE cells, which are derived from HMLE from prolonged treatment with TGF-β1 (Mani *et al*, 2008), were maintained in EMT-inducing media without additional TGF-β1. MCF-10A cells were cultured in DMEM:F12 media (1:1) supplemented with 10 μg/ml insulin, 20 ng/ml EGF, 0.5 μg/ml hydrocortisone, 100 ng/ml cholera toxin and 5% horse serum and induced to undergo EMT by adding 1.0 ng/ml of TGF-β1 for 14 days. MDCK cells were cultured in DMEM + 10% FCS and induced to undergo EMT by addition of 1.0 ng/ml of TGF-β1 for 12 days. SH-EP cells were maintained in DMEM + 10% FCS. LNCaP cells and derivative inducible ZEB1 (provided by Brett Hollier) and QKI-5 cells were maintained in RPMI + 10% FCS. For doxycycline induction experiments, 1 μg/ml was added to the media for 6 days followed by its removal for up to 10 days.

For transfection experiments, cells were transfected with 20 nM miRNA precursors (mirVana miRNA mimics; Ambion), 20 nM siRNA (ON-TARGETplus siRNA; Dharmacon) or 20–100 nM anti-miRNA (miRCURY LNA Power microRNA inhibitor; Exiqon) using Lipofectamine RNAiMAX (ThermoFisher) for 72 h unless otherwise specified prior to downstream functional assays. Details of miRNA precursors, anti-miRs and siRNA are provided in Table EV8. For the HMLE EMT timecourse, siRNAs were re-transfected upon passaging at each indicated time point.

Cell lines with stable overexpression or knockdown of QKI-5 were generated using vectors and methodology as previously described (Conn *et al*, 2015) and are listed in Table EV8. To generate cell lines with inducible QKI-5 expression, pENTR2B-QKI-5 were recombined with pInducer20 (Meerbrey *et al*, 2011) using LR Clonase to produce pInducer20-QKI-5. Lentivirus was produced from pInducer20-ZEB1 and pInducer20-QKI-5 following Lipofectamine 2000 (ThermoFisher)-mediated co-transfection of HEK293T cells with each vector along with pCMV-VSVG and pCMV-dR8.2. A 1:4 dilution of viral supernatant was used to transduce LNCaP and MCF-7 cell lines, after which pools of cells were selected for using 1.0 mg/ml of G418, respectively.

### RNA extraction and PCR

RNA was extracted using TRIzol (ThermoFisher) as per the manufacturer's instructions. For mRNA PCR, reverse transcription was carried out using random hexamers on 1.0 μg of total RNA using the QuantiTect RT kit (Qiagen) which was diluted 1:10 or 1:20 prior to quantitative PCR (qPCR) or semi-quantitative PCR (sqPCR). qPCR was performed in triplicate using the QuantiTect SYBR Green PCR kit (Qiagen) on a Rotor-Gene 6000 series PCR machine (Qiagen). Analysis was performed using comparative quantitation feature of the Rotor-Gene software with data normalised to GAPDH expression. sqPCR for alternative splicing analysis was performed using Standard Taq polymerase (NEB) or Phusion DNA Polymerase (ThermoFisher). MicroRNA qPCR was performed using 5.0 ng of RNA using TaqMan microRNA assay kits (Applied Biosystems) according to the manufacturer's instructions, with data normalised to U6 small nuclear RNA. For mapping of the QKI 3′UTRs, cDNA was synthesised using the specific reverse transcription primers (locations in Fig EV2) and sqPCR performed using Standard Taq polymerase (NEB) with QKI isoform-specific primer sets. All primer sequences are shown in Table EV8.

### Immunoblotting and quantitation

Protein extracts were prepared from transfected cells with Triton X-100 lysis buffer (50 mM HEPES, pH 7.5, 150 mM sodium chloride, 10 mM sodium pyrophosphate, 5 mM EDTA, 50 mM sodium fluoride, 1% Triton X-100 and protease inhibitor cocktail) and 20 μg fractionated on a SDS polyacrylamide gel. After transfer onto a nitrocellulose membrane and blocking in 5% skim milk for 1 h at room temperature, probing was carried in 5% skim milk overnight at 4°C using antibodies and dilutions as described in Table EV8. Membranes were exposed using the ECL method on a ChemiDoc imaging system (Bio-Rad). Quantitation relative to an α-tubulin loading control was carried out using FIJI software (Schindelin *et al*, 2012).

### Immunofluorescence and quantitation

Cells were transfected with microRNA precursors or siRNAs as described above, plated onto fibronectin-coated chamber slides (Nunc) and immunostained after 72 h. Cells were fixed in 4% paraformaldehyde, permeabilised in 0.1% Triton X-100 and probed with a 1:200 dilution of anti-E-cadherin, paxillin or β-catenin antibodies for 1 h at room temperature. Primary antibodies were detected with a 1:200 dilution of Alexa 488- and Alexa 594-conjugated secondary antibodies for 1 h at room temperature. For F-actin staining, fixed and permeabilised cells were incubated with 1:200 dilution of rhodamine phalloidin (Molecular Probes) for 30 min at room temperature. Sections were mounted in ProLong Gold Antifade Reagent containing DAPI (Molecular Probes). Confocal images were acquired under a confocal microscope (LSM 700; Zeiss). Antibodies are detailed in Table EV8.

For analysis of cell shape (length:width ratio) or cell area of F-actin-stained cells, measurements were made using the line tool on FIJI software (Schindelin *et al*, 2012). Cell length was defined by the longest distance between any two points of the cell, and cell width was measured as the longest line perpendicular to the cell-length line. Cell area was measured as the area encompassed following demarcation of the cell circumference. Measurements were conducted in a minimum of sixty F-actin-stained cells per treatment. For analysis of focal adhesion number and length, focal adhesions

were identified in whole paxillin-stained cells and measured with the arbitrary line tool on the AnalySIS LS software (Olympus). A minimum of 500 focal adhesions were counted on at least 20 cells per treatment.

### Luciferase reporter assays

An ~2.3-kb segment of the QKI-5 3′UTR was amplified from MDA-MB-231 gDNA and cloned into XhoI and NotI sites of psiCheck2 (Promega). Proximal (853 bp) and distal (443 bp) regions of the QKI-5 3′UTR containing miR-375 and miR-200c sites were amplified from psiCheck2-QKI-5 3′UTR and re-cloned into psiCheck2. Mutations in the miR-375 and miR-200c sites were generated by cloning in equivalent synthesised GeneArt gene fragments (ThermoFisher) to the proximal and distal regions. Transfections were carried out by co-transfecting 5 nM miRNA (mirVana miRNA mimics; Ambion) with psiCheck reporters in MDA-MB-231 cells in triplicate for 48 h. Reporter assays were performed three times using the Dual Luciferase reporter assay system (Promega) with one representative experiment shown. Primer sequences for cloning are shown in Table EV8.

### Migration and invasion assays

Migration or invasion assays were performed by plating cells in Transwells (Corning, 6.5 mm, 8.0 mm pore size) or BioCoat Matrigel chambers (BD Biosciences, 6.5 mm, 8.0 mm pore size), respectively, in serum-free medium with 0.01% bovine serum albumin. Chemotactic migration/invasion was induced be adding 10% FCS to the lower chamber. For MDA-MB-231 and mesHMLE cells, $2 \times 10^5$ or $8 \times 10^4$ cells were utilised in 4-h migration or 24-h invasion assays, respectively. For SH-EP cells, $8 \times 10^4$ cells were utilised in 24-h invasion experiments. Following experimental endpoint, membranes were fixed with 10% buffered formalin and cells stained with DAPI. Six fields of view for each membrane were counted using FIJI software (Schindelin et al, 2012). All experiments included at least 3–4 replicates and were performed multiple times with data pooled and shown as mean ± SEM.

For non-directional cell migration, transfected cells were plated at low density and images were collected on the IncuCyte Zoom (Essen Bioscience) every hour for 24 h. Cell tracking was performed manually with 25 cells per group using the manual tracking and chemotaxis tools on FIJI software (Schindelin et al, 2012).

### Colony formation assays

Soft agar colony formation assays were set up in 6-well plates using a base layer of 1.5 ml of 0.5% low-melting-point agarose (Sigma A9045) in normal growth media, with a top layer of 2.0 ml of 0.33% low-melting-point agarose/growth media incorporating $2 \times 10^4$ mesHMLE cells. Colonies were allowed to form for 14 days and were visualised by phase imaging. Colonies were enumerated using the particle counting module on FIJI software (Schindelin et al, 2012).

### Mice and tumour growth assays

NOD/SCID mice were housed in the SA Pathology Animal Care Facility, and experiments were conducted under the institutional

animal ethics guidelines. Mice were anaesthetised before injections of $1 \times 10^6$ MDA-MB-231 LM2 cells (Minn et al, 2005) in 50 ml of 50% Matrigel (BD Biosciences) into the fourth mammary gland. Ten mice per group were used for each cell line. Tumour volume was measured using electronic callipers every 3–5 days over a 33-day period.

### Clinical public data sets and merging with experimental data sets

To assess correlations between miR-200c or miR-375 and gene expression in clinical data sets, we performed linear regression analysis using R, generating Pearson correlation coefficients and q-values for all annotated genes. Clinical data sets used were Enerly breast cancer (GSE19783, $n = 101$) (Enerly et al, 2011), Taylor prostate cancer (GSE21032, $n = 111$) (Taylor et al, 2010), The Cancer Genome Atlas (TCGA) breast and prostate cancer ($n = 834$ and $466$, respectively) and the NCI-60 cancer cell line panel (Liu et al, 2010). Experimental data sets were merged with clinical data to generate a ranked list of most consistent miR-200c targets including: microarray data of untreated or MDCK cells treated with TGF-β for 12 days (MDCK EMT, GeneChip Canine 2.0 arrays (Affymetrix), this study), microarray data of untreated or HMLE cells treated with TGF-β for 15 days (HMLE EMT, GeneChip Human Gene 1.0 ST Array (Affymetrix), this study), microarray data of MDA-MB-231 transfected with negative control or miR-200b for 3 days (MB-231 + miR-200, GeneChip Human Gene 1.0 ST Array (Affymetrix), this study), and miR-200b targets identified by Ago-HITS-CLIP of miR-200b in MDA-MB-231 cells (miR-200 HITS-CLIP (Bracken et al, 2014)). Differential expression in microarray data sets was assessed using the Robust Multichip Average method in the Partek Genomics Suite (v6.5). Genes upregulated by more than 1.5-fold (MDCK EMT, $n = 1,467$) and 1.3-fold (HMLE EMT, $n = 2,690$) and downregulated by 1.5-fold (MB-231 + miR-200b, $n = 1,064$) which possessed conserved putative miR-200c target sites (TargetScan 6.2 (Garcia et al, 2011), $n = 1,057$) were included in the miR-200c target identification pipeline. Pearson correlations between miR-375 and QKI in the above clinical data sets, and relative expression of QKI in tumour subsets and significance calculations were assessed using GraphPad Prism software. The relative expression of QKI in prostate primary versus metastasis samples was assessed in the Oncomine platform (ThermoFisher). The relative expression of QKI in tumour versus normal samples in the panTCGA cancer panel was obtained from Firebrowse Gene Expression Viewer (http://firebrowse.org). Association recurrence of miR-200c with the 20 highest ranking miR-200c targets in the panTCGA, as well as QKI with all miRNAs, was assessed using the CancerMiner database (Jacobsen et al, 2013). Gene expression profiling of MDA-MB-231 cells transfected with negative control or QKI-5 siRNAs for 6 days was performed on GeneChip Human Gene 2.0 ST Array (Affymetrix). Microarray data for all data sets are provided in Table EV9.

### RNA-seq library preparation, gene expression and alternative splicing analysis

Poly(A) RNA was isolated using NEBNext Poly(A) mRNA Magnetic Isolation Module columns according to the manufacturer's instructions. The poly(A) fraction was size-fractionated and

concentrated using 3.53 AMPure RNAClean XP sample preparation (New England Biolabs) and eluted with water. Stranded RNA libraries were made using the NEBNext Ultra Directional RNA Library Prep Kit for Illumina (New England Biolabs), and 100-bp paired-end sequencing was performed on multiplexed libraries (between 2 and 4 indexes) on an Illumina HiSeq 2500 at the ACRF Cancer Genomics Facility. Three biological replicates were prepared for HMLE or MesHMLE and two replicates for knockdown of QKI or ectopic expression of miR-200c in MesHMLE cells. Details of read processing, mapping and statistics are provided in Table EV1 and described in detail in Appendix Supplementary Methods. Gene expression analysis was performed using the cuffdiff command from cufflinks v2.1.1 with genes having $\geq$ 2-fold change in expression with a *q*-value of $\leq 0.05$ considered significant. For alternative splicing analysis, to ensure robust results we utilised three diverse algorithms: rMATS (Shen *et al*, 2014), DEXSeq (Anders *et al*, 2012) and cuffdiff. A ranking system based on the integration of results from the three programs was used to shortlist genes for validation and is described in details in Appendix Supplementary Methods and presented in Table EV6. Figures were prepared from rMATS results, with genes possessing a percentage spliced in (PSI) difference of $\geq 5\%$ and a false discovery rate (FDR) of $\leq 0.05$ considered significant unless otherwise specified. Summary outputs of gene expression and alternative splicing analysis are provided in Tables EV2 and EV4. Further details on analysis of RNA-seq data are provided in Appendix Supplementary Methods.

### QKI-5-CLIP and sequencing

The QKI-CLIP method was adapted from published methods (Jensen & Darnell, 2008), incorporating modifications from eCLIP (Van Nostrand *et al*, 2016) and iCLIP (Sutandy *et al*, 2016) (Appendix Supplementary Methods; Appendix Fig S1). A detailed protocol is included in Appendix Supplementary Methods. In short, mesHMLE cells were crosslinked and QKI-5 bound RNA immunoprecipitated with a QKI-5-specific antibody (Bethyl, A300-183A). Libraries were prepared including a size-matched input control as per eCLIP (Van Nostrand *et al*, 2016). Peaks of reads signifying QKI-5 binding were identified using MACS2 (Zhang *et al*, 2008), and comparison of QKI-5 CLIP peaks and skipped exon events following QKI-5 knockdown or during EMT was performed using rMAPS (Park *et al*, 2016). Further details of QKI-CLIP analysis are provided in Appendix Supplementary Methods.

### Gene ontology analysis

Analysis of gene ontology was performed using the Functional Classification Tool of Database for Annotation, Visualization and Integrated Discovery (DAVID) (Huang da *et al*, 2009a,b). The most significant category from the top five most significant default annotation clusters is displayed for each analysis.

### TCGA splicing data analysis

To analyse the relationship between differential alternative splicing and QKI expression, The Cancer Genome Atlas (TCGA) gene expression and splice junction read count data were obtained from the

Broad GDAC Firehose website (http://gdac.broadinstitute.org/). Full details of the methods used are provided in Appendix Supplementary Methods. Briefly, a metric similar to PSI was calculated for every represented junction (in both orientations). Mean PSIs were calculated and compared for the 10% of samples with the highest and lowest expression of QKI. This was performed for seven epithelial-derived cancers (breast BRCA, prostate PRAD, bladder BLCA, pan-kidney KIPAN, lung adenocarcinoma LUAD, lung squamous cell carcinoma LUSC, stomach and esophageal STES), each of which comprised a minimum of 450 samples. The consistency of QKI-correlated changes in splicing was analysed by unsupervised clustering, where the genes represented were constitutively expressed in all cancers and possess large QKI-correlated changes in splicing (a mean difference in PSI between QKI-high and QKI-low samples of at least 35% in at least one cancer). Further details are provided in Appendix Supplementary Methods.

### Statistics and reproducibility

Statistical analysis was carried out using GraphPad Prism software and data are represented as the mean $\pm$ SD unless otherwise indicated. For migration, invasion and cell tracking assays, results are shown as shown as mean $\pm$ SEM with significance measured by two-tailed unpaired *t*-tests. For measurement of cell shape, area and focal adhesion length, significance was measured by Mann–Whitney test. For comparison of cancer subtypes, significance was measured by two-tailed unpaired *t*-tests. Significance is expressed as *$P < 0.05$, **$P < 0.01$ and ***$P < 0.001$.

### Data availability

The RNA sequencing data from this publication have been deposited to the European Nucleotide Archive database (http://www.ebi.ac.uk/ena/data/view/PRJEB25042) with the study accession number PRJEB25042. The QKI-5 HITS-CLIP data from this publication have been deposited in NCBI's Gene Expression Omnibus (https://www.ncbi.nlm.nih.gov/geo/query/acc.cgi?acc = GSE111188) and are accessible through GEO Series accession number GSE111188.

**Expanded View** for this article is available online.

## Acknowledgements

We thank all members of the Gregory and Goodall laboratories for important inputs. This work was supported by grants from the National Health and Medical Research Council of Australia to P.A.G and G.J.G (GNT1068773 and GNT1128479) to L.A.S., W.D.T., G.J.G., B.G.H. and P.A.G. (GNT1083961). P.A.G. was supported by a Cancer Council of South Australia Beat Cancer Fellowship. L.A.S. was supported by a Young Investigator Award from the Prostate Cancer Foundation (Foundation 14 award). The results published here are in part based on data generated by The Cancer Genome Atlas, established by the National Cancer Institute and the National Human Genome Research Institute, and we are grateful to the specimen donors and relevant research groups associated with this project.

## Author contributions

PAG and GJG designed the study. KAP, JT, DML and LAS carried out bioinformatics analysis. AWS supervised bioinformatics analysis. PAG, CAP, SR, BKD, AGB, RL, DPN, XL, SJC and DL carried out experiments. CPB, BGH, NS, WDT,

YK-G and LAS provided reagents and intellectual input. PAG and GJG supervised the study and wrote the manuscript, with input from the other authors.

## Conflict of interest

The authors declare that they have no conflict of interest.

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
