## [Review Process File · The EMBO Journal]

miR-200/375 control epithelial plasticity-associated alternative splicing by repressing the RNA-binding protein Quaking

Katherine A. Pillman, Caroline A. Phillips, Suraya Roslan, John Toubia, B. Kate Dredge, Andrew G. Bert, Rachael Lumb, Daniel P. Neumann, Xiaochun Li, Simon J. Conn, Dawei Liu, Cameron P. Bracken, David M. Lawrence, Nataly Stylianou, Andreas W. Schreiber, Wayne D. Tilley, Brett G. Hollier, Yeesim Khew-Goodall, Luke A. Selth, Gregory J. Goodall and Philip A. Gregory.

Review timeline:

Submission date:	12 th January 2018
Editorial Decision:	9 th February 2018
Revision received:	20 th February 2018
Editorial Decision:	16 th March 2018
Revision received:	22 nd March 2018
Accepted:	24 th March 2018

Editor: Anne Nielsen

Transaction Report:

1st Editorial Decision

9th February 2018

Thank you for submitting your manuscript for consideration by The EMBO Journal. It has now been seen by three referees whose comments are shown below.

As you will see from the reports, our referees all express interest in the findings reported in your manuscript although they also ask for several aspects of the study to be strengthened before a revised version can be published in The EMBO Journal.

Given the referees' positive recommendations, I would like to invite you to submit a revised version of the manuscript, addressing the comments of all three reviewers. I should add that it is EMBO Journal policy to allow only a single round of revision, and acceptance of your manuscript will therefore depend on the completeness of your responses in this revised version.

For the revised manuscript I would particularly ask you to focus on the following points:

- > Please elaborate on the relevance for human cancer - if possible - as suggested by ref #1. As the referee outlines, this may be done via database searches and discussion rather than the inclusion of new primary data per se
- > Please make sure that information on statistics and number of repeats is given whenever relevant (refs #1 and #3)
- > In addition, please also discuss the role for QKI in EMT in the context of additional splice factors linked to the process (ref #2)
- > Refs #2 and #3 both ask you to include data on QKI overexpression to test its contribution as an EMT driver and I would encourage you to follow that suggestion.

 REFEREE REPORTS

Referee #1:

In the manuscript by Pillman et al., they provide convincing evidence that QKI-5 plays a critical role as a regulator of miR-200c-driven mRNA splicing and actin cytoskeletal remodeling. They identified QKI-5 as a miR-200c target by a ranking method that merges miRNA-mRNA correlation, EMT data sets, and AGO-HITS-CLIP data. They found that QKI-5 3'UTR is directly targeted by miR-200c and miR-375, and its expression is tightly controlled by these two miRNAs during EMT. Furthermore, they showed that QKI-5 regulates cancer cell morphology, promotes cell migration/invasion, and inhibits colony formation and tumor growth. They found QKI-5 regulates alternative splicing without changing gene expression levels, and identified a number of potential QKI-5 binding sites by HITS-CLIP. By the TCGA data mining they found QKI-regulated alternative splicing events are strongly correlated with QKI levels. Finally, they showed that miR-200c regulates gene expression by directly targeting and controlling alternative splicing of actin cytoskeleton associated genes through repressing QKI-5. Overall, the manuscript is clearly written. The findings are interesting and the conclusions are nicely supported by the rich data. Particularly interesting is the finding that miR-200c coordinately regulates mRNA levels and splicing through QKI-5.

There are minor concerns.

1. The data suggest that QKI-5 has an ambiguous role in cancer progression, promoting migration and invasion but inhibiting colony formation and tumor growth. Is QKI-5 prognostic in human cancers? Is QKI5-regulated alternative splicing signature prognostic in human cancers?
2. In the last part of the manuscript the authors stated that miR-200c coordinately regulates gene expression and alternative splicing of actin cytoskeleton associated genes. Did alternative splicing affect the functions of these genes? The authors should discuss this point.
3. A reference is missing in the first paragraph in Materials and methods section.
4. Figure 2, 3 and 4c lack statistical information and p-values.

Referee #2:

The manuscript by Pillman et al provides a detailed analysis of the role of the mir-200 family, mir-375, and the splicing factor Quaking (QKI) in the regulation of large scale changes in splicing during the epithelial to mesenchymal transition (EMT). While the precise role of EMT in various disease states, including cancer is increasingly debated in recent years, there is no doubt that this important developmental process can be hijacked in some cancer states. As such this study will be of interest to numerous investigators in the cancer field, EMT field, as well as the broad field of RNA biology. The study is performed by an excellent group of researchers who were among the first to identify the mir-200 family of microRNAs as epithelial-specific microRNAs with important functions in EMT. Through analysis of cell line data and from that of model EMT systems, they identify QKI as being highly anti-correlated with microRNA-200 members as well as mir-375. It is noted however, that previous groups have also shown a role of QKI in regulation of splicing in EMT and other cancer cell models. However, they provide convincing evidence that mir-200c and mir-375 directly target QKI-5 to decrease its expression at the RNA level as well as the protein level. They also show that QKI expression levels decrease (reversibly) in several model systems of EMT and depletion of QKI in mesenchymal cell types is associated with some changes in cell shape and function similar to reversion to an epithelial cell state, although without reversion of standard EMT markers. However, the most in depth and comprehensive analysis uses the well-traveled HMLE human breast cancer EMT model to conduct a genome-wide analysis of gene expression changes and splicing in response to TGF-beta and the corresponding effects of QKI depletion or mir-200c expression in "mesHMLE" cells using high depth RNA-Seq analysis. At the total gene expression level, mir-200 is largely able to revert gene expression changes back to the epithelial cell state, whereas depletion of QKI by itself is not, including that of standard epithelial markers. However, at the level of splicing, depletion of QKI as well as expression of mir-200c both induce splicing

patterns back to an epithelial pattern. A good number of validations are performed along with a comparison to splicing patterns in a panel of epithelial and mesenchymal cell lines. They also carry out a HITS-CLIP analysis to show that very many changes observed in response to modulation of QKI levels are in fact direct targets. These results are used to argue for a mir-200-QKI axis that plays a prominent role in splicing during EMT. As a whole the study is very well performed by an outstanding group of investigators. The data is generally well presented and interpretable and will be of great interest to many researchers. I do however, have some reservations regarding the study, including the strong degree to which it is proposed that changes in QKI expression alone are "necessary and sufficient" to account for most of the major changes in splicing that occur during EMT. There are ample reports from a number of previous investigations showing central roles for other splicing factors, including ESRPs and RBFOX family members in regulation of splicing during EMT. While these are mostly referenced in the text, the combinatorial roles of other splicing factors in regulation of splicing in EMT are minimized with an overly strong emphasis on the role of mir-200c in regulating QKI expression alone in achieving these splicing changes. It is noted, for example that several of the EMT associated alternative exons the authors validate as QKI targets were also previously shown to be regulated by the ESRPs. A more nuanced and complete accounting of the role of some of these other factors in the model for EMT splicing changes should be better outlined by the authors. The authors should present a more complete model to indicate how changes in levels of mir-200 and QKI as well as in other factors cooperate to achieve splicing changes in EMT. Specific critiques are outlined below, largely in the order in which they arise in the manuscript.

- There is some confusion/lack of clarity in Fig 2b. The data showing that both mir-200c and mir-375 downregulate the reporter with the full length QKI-5 UTR, it is not clear how the proximal and distal UTRs were introduced into the reporter (? Just the 5' and 3' ends alone, or just by deleting the other miR target sites). It would be helpful to expand the schematic to illustrate this more clearly and also to show the specific sites and the mutations used. The figure as stands is small, so this should be easily accomplished. It is also not clear nor discussed what is going on in the last two sets of overexpression where there are no appreciable changes in reporter activity (e.g., if the mir375 sites alone are mutated, why is the reporter not responsive to mir-200c (and vice versa).
- Page 6 "Overexpression of miR-200c and miR-375 each reduced the level of QKI-5 and QKI-6, and strongly decreased all 3 QKI isoform proteins" is not fully supported in Fig. 2A. There does not appear to be appreciable reduction in QKI-7 with miR-200c.
- Small issue, page 8 last sentence of paragraph one. The relationship between QKI OE or siQKI and reversion to epithelial cell phenotype is unclear. While both were done, it would seem they mean that depletion of QKI does not revert to epithelial cell state.
- The splicing analysis by RNA-Seq and the validations by RT-PCR are generally well done. However, the authors should also indicate the percent exon inclusion values for each RT-PCR to give a clearer indication of the degree of splicing changes observed. This could be done either by including the values below each lane, or possibly in a graph.
- Related to the primary issue regarding the role of other splicing factors. The authors state that "QKI is necessary (or) and sufficient to regulate these splicing events and conclude that "Together these data demonstrate that QKI-5 is the primary target by which these miRNAs regulate EMT-associated alternative splicing events." However, as the authors note, Zeb1 is also a direct mir-200c target and several studies have shown direct repression of the ESRPs by ZEB1 during EMT, where it is nearly completely downregulated. It is noted that in fact, 4 of the 11 alternative exons validated by the authors (ADD3, NUMB, CLSTN1, and NFYA) were also previously shown to switch splicing towards the mesenchymal pattern in response to ESRP depletion (Yang et al 2016-cited, but these relevant observations are not noted here). Indeed this prior study showed combinatorial regulation of the ADD3 and CLSTN1 alternative exons by both ESRPs and QKI-5. The authors would need to provide some discussion regarding this point. Is it possible that both the expression of ESRPs and QKI depletion have some redundancy in terms of regulating epithelial-mesenchymal splicing? It is suggested that the authors also test ESRP OE in mesHMLE cells and or depletion in HMLE to determine the combinatorial nature of splicing regulation in this EMT model. While QKI OE is shown to somewhat augment these patterns in mesHMLE, a stronger argument for primacy of QKI in regulating these events would be to ectopically express QKI-5 in HMLE cells that have not been induced to undergo EMT. The inducible expression of QKI-5 in LNCAP and MCF7 cells seems to have a less robust effect of splicing of several of these targets (e.g. NFYA, MYO18A, CD47) suggesting that it alone may not be sufficient to fully enforce mesenchymal splicing in the context of normal epithelial cells. In order to support the "sufficient" argument, the authors would need to perform QKI OE in HMLE cells since this was the primary EMT system used and would

provide a more complete determination as to whether it is fully sufficient to convert splicing to the mesenchymal pattern. This would also require complete quantification of these splicing changes using percent exon inclusion. It would seem likely that induction of QKI in EMT works in concert with other splicing factors implicated in this process to achieve the full degree of splicing change in many of these targets.

- Related to the above in discussion: "Comparison of our data with this earlier study indicates that although QKI and ESRP1/2 co-regulate a subset of events in opposing directions, these activities are largely distinct." This statement illustrates a general problem associated in using Venn diagrams to draw these types of conclusions from RNA-Seq analysis, especially for splicing, given a large false negative problem. First, the authors used a much higher depth of RNA-Seq than the prior study and thus, not surprisingly, identified a larger set of splicing changes. Second, the prior study was performed in a different cell line and so the results cannot be directly compared, at least if such a strong conclusion is to be arrived at. Third, the study used to indicate this lack of overlap does not account for a larger number of ESRP targets identified in several other studies, including those identified in mouse *Esrp* KO studies. To support this strong claim, the authors would need to provide data from a panel of QKI regulated and EMT-associated splicing targets (with appreciable changes in present exon splicing) and show that depletion of ESRPs in the HMLE cells (without EMT induction) does not impact splicing and possibly also that introducing ESRP expression into mesHMLE cells after QKI depletion is unable to provide any reversion to an epithelial splicing pattern (as noted above).

- The statement that previous "studies have indicated QKI is involved in cancer associated splicing, its action to date has largely been inferred from motif analysis or from effects on single target genes" is a bit strong. The paper cited by Zong et al did use RNA-Seq to carry out a genome wide analysis of QKI splicing in response to QKI depletion by RNA-Seq, albeit at a lower depth of sequencing and identification of a correspondingly smaller number of regulated splicing targets for QKI.

- In the abstract "...mediate miRNA induced changes in splicing" while not technically inaccurate is a bit confusing and might be interpreted to mean a direct effect. Perhaps the language could be changed to indicate more clearly that these changes are proposed to be indirect to help the general reader.

Referee #3:

In their manuscript Pillman and colleagues describe the impact of miR-200-QKI RBP on alternative splicing during EMT. They demonstrate this association with cancer datasets, and show the relevance for changes in actin cytoskeleton-associated genes, cell migration and invasion. Altogether, the authors present a very strong manuscript, controlled experiments, and a very interesting topic.

Comments:

Figure 2. Data should include pValue and statistics of the relevant comparisons

Figure 4D, it is strange that shQKI#5 in MDA-231 cells has little effect on protein levels, while strongly impacting migration and invasion. This shRNA vector targets the 3'UTR of QKI-5 and therefore may have less activity. I suggest to replace this experiment with a more convincing one.

Figure 4G: I miss the mice with QKI over-expression to complete the picture. I wonder if the authors have already tried it and if it had any effect at all.

Figure 5E: The authors show QKI-5 HITS-CLIP results identifying the motif ACUAAC. They show that exon skipping events were associated with QKI binding within or upstream of the skipped 3' splice site. In contrast, exon inclusion involved binding downstream. This is a very interesting and relevant result, but only association. To strengthen it, the authors should examine by reporter assays or in vitro that indeed the motif sequences at the suggested places impact exon skipping/inclusion frequencies.

Re: EMBOJ-2018-99016

We thank the reviewers for their thoughtful comments and suggestions for improvement. As recommended by the editor, we have focused our revision on the four suggested points but have also addressed the other points raised by the reviewers.

For the revised manuscript I would particularly ask you to focus on the following points:

-> Please elaborate on the relevance for human cancer - if possible - as suggested by ref #1. As the referee outlines, this may be done via database searches and discussion rather than the inclusion of new primary data per se

- We have performed database searches looking for associations of QKI with clinical outcome in cancer datasets. We find in terms of patient outcome in breast cancer, that high QKI levels are indicative of poor distant metastasis-free survival in unsegregated combined cohort data from KM plotter (n=1746) and the Hatzis et al (n=508) datasets. We have included these extra data in Figure EV1C and added the underlined text “Examination of QKI expression in breast cancer cohorts showed it was upregulated in basal-like and claudin-low subtypes, which display enhanced EMT-like features, and is indicative of poor distant metastasis -free survival (Fig. 1D and Fig. EV1B and C)” on page 5. A description of the methodology used to generate these graphs are in Appendix Supplementary Methods.

-> Please make sure that information on statistics and number of repeats is given whenever relevant (refs #1 and #3)

- We have added information on the statistics used and numbers of biological repeats to the figure legends 2 and 4 where relevant. In some cases, the data shown are technical qPCR replicates and adding statistics is not relevant. In these cases we have removed error bars. For Figure 2B, we show a biological triplicate experiment and thus have added significance testing to this figure. For Figure 4C (number of focal adhesions per cell), we recounted individual discrete cells (at least 20 cells per treatment) to obtain a measure of the standard deviation and have replaced the original graph with this new data (with actual numbers of cells counted indicated in the figure legend). Where appropriate biological replicates have been carried out, we have shown the statistical significance and clarified the number of samples and the statistical test used in the figure legend.

-> In addition, please also discuss the role for QKI in EMT in the context of additional splice factors linked to the process (ref #2)

- We have expanded our discussion on the interaction of ESRP and QKI in splicing as reflected in the comments to reviewer 2 below.

-> Refs #2 and #3 both ask you to include data on QKI overexpression to test its contribution as an EMT driver and I would encourage you to follow that suggestion.

- We have over-expressed QKI-5 in HMLE cells and similar to MCF-7 and LNCaP cells, we find that QKI-5 is sufficient to cause EMT splicing changes but does so in the absence of driving EMT – as indicated by a lack of change in cell morphology and EMT marker expression. Furthermore, as suggested by reviewer #2 we calculated change in PSI on this dataset. We have added these data as Figure EV5D and this is now referred to in the text on page 11. A description of the methodology used to generate this data is included in Appendix Supplementary Methods.

Referee #1:

In the manuscript by Pillman et al., they provide convincing evidence that QKI-5 plays a critical role as a regulator of miR-200c-driven mRNA splicing and actin cytoskeletal remodeling. They identified QKI-5 as a miR-200c target by a ranking method that merges miRNA-mRNA correlation, EMT data sets, and AGO-HITS-CLIP data. They found that QKI-5 3'UTR is directly targeted by miR-200c and miR-375, and its expression is tightly controlled by these two miRNAs during EMT. Furthermore, they showed that QKI-5 regulates cancer cell morphology, promotes cell migration/invasion, and inhibits colony formation and tumor growth. They found QKI-5 regulates alternative splicing without changing gene expression levels, and identified a number of potential QKI-5 binding sites by HITS-CLIP. By the TCGA data mining they found QKI-regulated alternative splicing events are strongly correlated with QKI levels. Finally, they showed that miR-200c regulates gene expression by directly targeting and controlling alternative splicing of actin cytoskeleton associated genes through repressing QKI-5. Overall, the manuscript is clearly written. The findings are interesting and the conclusions are nicely supported by the rich data. Particularly interesting is the finding that miR-200c coordinately regulates mRNA levels and splicing through QKI-5.

There are minor concerns.

1. The data suggest that QKI-5 has an ambiguous role in cancer progression, promoting migration and invasion but inhibiting colony formation and tumor growth. Is QKI-5 prognostic in human cancers? Is QKI5-regulated alternative splicing signature prognostic in human cancers?

- We have performed database searches looking for associations of QKI with clinical outcome as described in our response to the Editor's main point above. Assessing whether QKI5-regulated alternative splicing signature is also prognostic for distant metastasis-free survival would require availability of raw primary sequence data linked to individual patient outcomes, which we are not aware of.

2. In the last part of the manuscript the authors stated that miR-200c coordinately regulates gene expression and alternative splicing of actin cytoskeleton associated genes. Did alternative splicing affect the functions of these genes? The authors should discuss this point.

- We have added an extra sentence related to the published function of the several targets in the final paragraph of the discussion on page 17 as follows "Several of these alternative spliced products including ENAH (MENA), PARD3, and NIN display altered phosphorylation, protein-protein interactions, and cell localisation respectively (Balsamo, Mondal et al., 2016, Gao, Macara et al., 2002, Zhang, Chen et al., 2016) "

3. A reference is missing in the first paragraph in Materials and methods section.

- We have added this reference.

4. Figure 2, 3 and 4c lack statistical information and p-values.

- We have addressed this in response to the editor's main points above. Statistical information has been added where applicable.

Referee #2:

The manuscript by Pillman et al provides a detailed analysis of the role of the mir-200 family, mir-375, and the splicing factor Quaking (QKI) in the regulation of large scale changes in splicing during the epithelial to mesenchymal transition (EMT). While the precise role of EMT in various disease states, including cancer is increasingly debated in recent years, there is no doubt that this important developmental process can be hijacked in some cancer states. As such this study will be of interest to numerous investigators in the cancer field, EMT field, as well as the broad field of RNA biology. The study is performed by an excellent group of researchers who were among the first to identify the mir-200 family of microRNAs as epithelial-specific microRNAs with important functions in EMT. Through analysis of cell line data and from that of model EMT systems, they identify QKI as being highly anti-correlated with microRNA-200 members as well as mir-375. It is

noted however, that previous groups have also shown a role of QKI in regulation of splicing in EMT and other cancer cell models. However, they provide convincing evidence that mir-200c and mir-375 directly target QKI-5 to decrease its expression at the RNA level as well as the protein level. They also show that QKI expression levels decrease (reversibly) in several model systems of EMT and depletion of QKI in mesenchymal cell types is associated with some changes in cell shape and function similar to reversion to an epithelial cell state, although without reversion of standard EMT markers. However, the most in depth and comprehensive analysis uses the well-traveled HMLE human breast cancer EMT model to conduct a genome-wide analysis of gene expression changes and splicing in response to TGF-beta and the corresponding effects of QKI depletion or mir-200c expression in "mesHMLE" cells using high depth RNA-Seq analysis. At the total gene expression level, mir-200 is largely able to revert gene expression changes back to the epithelial cell state, whereas depletion of QKI by itself is not, including that of standard epithelial markers. However, at the level of splicing, depletion of QKI as well as expression of mir-200c both induce splicing patterns back to an epithelial pattern. A good number of validations are performed along with a comparison to splicing patterns in a panel of epithelial and mesenchymal cell lines. They also carry out a HITS-CLIP analysis to show that very many changes observed in response to modulation of QKI levels are in fact direct targets. These results are used to argue for a mir-200-QKI axis that plays a prominent role in splicing during EMT. As a whole the study is very well performed by an outstanding group of investigators. The data is generally well presented and interpretable and will be of great interest to many researchers. I do however, have some reservations regarding the study, including the strong degree to which it is proposed that changes in QKI expression alone are "necessary and sufficient" to account for most of the major changes in splicing that occur during EMT. There are ample reports from a number of previous investigations showing central roles for other splicing factors, including ESRPs and RBFOX family members in regulation of splicing during EMT. While these are mostly referenced in the text, the combinatorial roles of other splicing factors in regulation of splicing in EMT are minimized with an overly strong emphasis on the role of mir-200c in regulating QKI expression alone in achieving these splicing changes. It is noted, for example that several of the EMT associated alternative exons the authors validate as QKI targets were also previously shown to be regulated by the ESRPs. A more nuanced and complete accounting of the role of some of these other factors in the model for EMT splicing changes should be better outlined by the authors. The authors should present a more complete model to indicate how changes in levels of mir-200 and QKI as well as in other factors cooperate to achieve splicing changes in EMT. Specific critiques are outlined below, largely in the order in which they arise in the manuscript.

- There is some confusion/lack of clarity in Fig 2b. The data showing that both mir-200c and mir-375 downregulate the reporter with the full length QKI-5 UTR, it is not clear how the proximal and distal UTRs were introduced into the reporter (? Just the 5' and 3' ends alone, or just by deleting the other miR target sites). It would be helpful to expand the schematic to illustrate this more clearly and also to show the specific sites and the mutations used. The figure as stands is small, so this should be easily accomplished. It is also not clear nor discussed what is going on in the last two sets of overexpression where there are no appreciable changes in reporter activity (e.g., if the mir375 sites alone are mutated, why is the reporter not responsive to mir-200c (and vice versa).

- **To clarify we expanded this illustration to clearly delineate the proximal and distal region constructs used in the reporter assays. We changed the text on page 6 to read "Analysing the QKI-5 3'UTR proximal and distal regions in isolation" instead of "Dividing the QKI-5 3'UTR into proximal and distal regions" for further clarity (see page 6).**

- Page 6 "Overexpression of miR-200c and miR-375 each reduced the level of QKI-5 and QKI-6, and strongly decreased all 3 QKI isoform proteins" is not fully supported in Fig. 2A. There does not appear to be appreciable reduction in QKI-7 with miR-200c.

- **We have modified this sentence, removing the word "strongly", to account for the more modest decrease of QKI-7 by miR-200c in mesHMLE cells.**

- Small issue, page 8 last sentence of paragraph one. The relationship between QKI OE or siQKI and reversion to epithelial cell phenotype is unclear. While both were done, it would seem they mean that depletion of QKI does not revert to epithelial cell state.

- To remove this ambiguity, we have modified this sentence to “Together, these data demonstrate QKI-5 stimulates changes in cell migration, invasion, focal adhesions, and morphology, but unlike miR-200c, knockdown of QKI-5 does so without reverting cells to a fully epithelial phenotype”.

- The splicing analysis by RNA-Seq and the validations by RT-PCR are generally well done. However, the authors should also indicate the percent exon inclusion values for each RT-PCR to give a clearer indication of the degree of splicing changes observed. This could be done either by including the values below each lane, or possibly in a graph.

- We have provided graphs of change in PSI for several representative events as measured by qPCR for the HMLE + TGF- β timecourse in Figures 6A (shown as Figure EV5C) and the HMLE-iQKI-5 timecourse (shown as Figure EV5D). We feel adding extra values to the figures would obscure rather than enhance the findings presented, especially given there are such a large number of panels to which such splicing numbers would need to be added, and for many events the change in splicing is clearly evident. The PSI changes calculated by rMATS analysis of RNA-seq data are shown in Appendix Table S4.

- Related to the primary issue regarding the role of other splicing factors. The authors state that "QKI is necessary (or) and sufficient to regulate these splicing events and conclude that "Together these data demonstrate that QKI-5 is the primary target by which these miRNAs regulate EMT-associated alternative splicing events." However, as the authors note, Zeb1 is also a direct miR-200c target and several studies have shown direct repression of the ESRPs by ZEB1 during EMT, where it is nearly completely downregulated. It is noted that in fact, 4 of the 11 alternative exons validated by the authors (ADD3, NUMB, CLSTN1, and NFYA) were also previously shown to switch splicing towards the mesenchymal pattern in response to ESRP depletion (Yang et al 2016-cited, but these relevant observations are not noted here). Indeed this prior study showed combinatorial regulation of the ADD3 and CLSTN1 alternative exons by both ESRPs and QKI-5. The authors would need to provide some discussion regarding this point. Is it possible that both the expression of ESRPs and QKI depletion have some redundancy in terms of regulating epithelial-mesenchymal splicing? It is suggested that the authors also test ESRP OE in mesHMLE cells and or depletion in HMLE to determine the combinatorial nature of splicing regulation in this EMT model. While QKI OE is shown to somewhat augment these patterns in mesHMLE, a stronger argument for primacy of QKI in regulating these events would be to ectopically express QKI-5 in HMLE cells that have not been induced to undergo EMT. The inducible expression of QKI-5 in LNCAP and MCF7 cells seems to have a less robust effect of splicing of several of these targets (e.g. NFYA, MYO18A, CD47) suggesting that it alone may not be sufficient to fully enforce mesenchymal splicing in the context of normal epithelial cells. In order to support the "sufficient" argument, the authors would need to perform QKI OE in HMLE cells since this was the primary EMT system used and would provide a more complete determination as to whether it is fully sufficient to convert splicing to the mesenchymal pattern. This would also require complete quantification of these splicing changes using percent exon inclusion. It would seem likely that induction of QKI in EMT works in concert with other splicing factors implicated in this process to achieve the full degree of splicing change in many of these targets.

- We have performed QKI-5 overexpression in HMLE cells as suggested and found that QKI-5 is indeed sufficient to effect mesenchymal splicing in this context and in the absence of effects on EMT (see response to editor's main points above). We have added this data as Figure EV5D and provide qPCR quantitation of change in PSI for several representative events. Of note, the degree of PSI change is similar to that observed after TGF- β induced EMT of HMLE cells (see Figure EV5C). We agree that QKI works in concert with other splicing factors and in particular ESRP1. Our understanding of reading the cited Yang et al 2016 paper is that ADD3 was not regulated by ESRP1/2 kd (see Fig 3C within) while CLSTN1 is regulated in opposing directions by ESRP and QKI. CLSTN1 has been highlighted in Appendix Figure S1B (and see next question for resulting changes made to text)

- Related to the above in discussion: "Comparison of our data with this earlier study indicates that

although QKI and ESRP1/2 co-regulate a subset of events in opposing directions, these activities are largely distinct." This statement illustrates a general problem associated in using Venn diagrams to draw these types of conclusions from RNA-Seq analysis, especially for splicing, given a large false negative problem. First, the authors used a much higher depth of RNA-Seq than the prior study and thus, not surprisingly, identified a larger set of splicing changes. Second, the prior study was performed in a different cell line and so the results cannot be directly compared, at least if such a strong conclusion is to be arrived at. Third, the study used to indicate this lack of overlap does not account for a larger number of ESRP targets identified in several other studies, including those identified in mouse ESRP KO studies. To support this strong claim, the authors would need to provide data from a panel of QKI regulated and EMT-associated splicing targets (with appreciable changes in present exon splicing) and show that depletion of ESRPs in the HMLE cells (without EMT induction) does not impact splicing and possibly also that introducing ESRP expression into mesHMLE cells after QKI depletion is unable to provide any reversion to an epithelial splicing pattern (as noted above).

- Taken with the above commentary we have modified this section to reflect a more nuanced conclusion incorporating specific information with the Yang et al paper. It now reads "A recent study indicated ESRP1/2 and QKI could co-regulate and induce opposing alternative splicing in at least twelve splice events, including CLSTN1, MPRIP, EXOC1, and DEPDC1 which we describe here to be QKI regulated (Yang et al., 2016). Comparison of our data with this earlier study carried out in a different cell line indicates that although QKI and ESRP1/2 co-regulate a subset of events in opposing directions, their activities appear to be largely distinct (Appendix Fig. S1B)"

- The statement that previous "studies have indicated QKI is involved in cancer associated splicing, its action to date has largely been inferred from motif analysis or from effects on single target genes" is a bit strong. The paper cited by Zong et al did use RNA-Seq to carry out a genome wide analysis of QKI splicing in response to QKI depletion by RNA-Seq, albeit at a lower depth of sequencing and identification of a correspondingly smaller number of regulated splicing targets for QKI.

- We have modified this sentence to "Although several studies have indicated QKI is involved in cancer-associated splicing, its action to date has largely been inferred from motif analysis or from effects on single target genes, with the exception of a global analysis in lung cancer cells)" to specifically highlight the Zong et al paper mentioned.

- In the abstract "...mediate miRNA induced changes in splicing" while not technically inaccurate is a bit confusing and might be interpreted to mean a direct effect. Perhaps the language could be changed to indicate more clearly that these changes are proposed to be indirect to help the general reader.

- We have modified this sentence to "This is achieved by their strong suppression of the RNA binding protein Quaking (QKI), which is required to mediate the splicing changes regulated by these miRNAs" to avoid confusion.

Referee #3:

In their manuscript Pillman and colleagues describe the impact of miR-200-QKI RBP on alternative splicing during EMT. They demonstrate this association with cancer datasets, and show the relevance for changes in actin cytoskeleton-associated genes, cell migration and invasion. Altogether, the authors present a very strong manuscript, controlled experiments, and a very interesting topic.

Comments:

Figure 2. Data should include pValue and statistics of the relevant comparisons

- We have addressed this as described in the response to the editor's main points above.

Figure 4D, it is strange that shQKI#5 in MDA-231 cells has little effect on protein levels, while

strongly impacting migration and invasion. This shRNA vector targets the 3'UTR of QKI-5 and therefore may have less activity. I suggest to replace this experiment with a more convincing one.

- **shQKI5 is less potent than shQKI2 in MDA-MB-231 cells and the degree of reduction of migration and invasion correlate with the level of repression, strengthening the validity of our conclusion.**

Figure 4G: I miss the mice with QKI over-expression to complete the picture. I wonder if the authors have already tried it and if it had any effect at all.

- **We have not yet tried overexpressing QKI-5 in MDA-MB-231 LM2 cells and performing mouse experiments.**

Figure 5E: The authors show QKI-5 HITS-CLIP results identifying the motif ACUAAC. They show that exon skipping events were associated with QKI binding within or upstream of the skipped 3' splice site. In contrast, exon inclusion involved binding downstream. This is a very interesting and relevant result, but only association. To strengthen it, the authors should examine by reporter assays or in vitro that indeed the motif sequences at the suggested places impact exon skipping/inclusion frequencies.

- **We agree this is only an association. We will carry out a more detailed analysis and present this in a subsequent manuscript.**

2nd Editorial Decision

16th March 2018

Thank you for submitting a revised version of your manuscript. It has now been seen by two of the original referees whose comments are shown below.

As you will see, ref #3 is satisfied with the revision while ref #2 raises a remaining concern that will have to be addressed either with the inclusion of additional data/analysis or with a moderation of the conclusion of the overlap between ESRP and QKI-5. I would therefore like to invite you to submit a final revision of the manuscript in which you address this issue.

REFeree REPORTS

Referee #2:

The authors have responded to most of my critiques, but the response to one of my major critiques is inadequately addressed. This relates to the degree of overlap or combinatorial regulation of EMT splicing switches by QKI and ESRPs. The response does not address several of the main points of the critique and they basically do not change their conclusion by merely adding "appear to be" largely distinct subsets of alternative splicing events. The authors do not list ADD3 and NUMB as ESRP regulated events, I believe, by misinterpreting the data in the Yang paper. These EMT associated events, while not passing the statistical thresholds as regulated events in ESRP depleted cells by RNA-Seq, nonetheless were confirmed to be regulated by them by RT-PCR. For ADD3, the changes in splicing was relatively small, but still statistically significant. Such examples illustrate the false negative issue for splicing analysis in RNA-Seq datasets that was pointed out in the critique. The authors also did not consider or compare to other ESRP depletion datasets or those of orthologous events from RNA-Seq analysis of mouse ESRP1/2 KO mice as suggested in the critique. Thus, of the 11 QKI validated events they validated, in fact 8 of the 11 have been shown to also be regulated by ESRPs/Esprs (ADD3, DEPDC1, EXOC1, NUMB, CLSTN1, MPRIP, MYO18A, and NYFA). The authors ought to point out all of these co-regulated events. Based on these validated examples the statement that their activities appear to be largely distinct is not supported. If the authors wish to retain this claim, they would need to carry out a larger analysis of regulated by both factors in the same cell system as suggested in the critique. Alternatively, they could modify the statement to simply state that the degree of overlap in co-regulated targets would need to be determined by global splicing analysis in the same cell type(s).

Referee #3:

The revised manuscript addressed all my concerns.

2nd Revision - authors' response

22nd March 2018

Re: EMBOJ-2018-99016R1

Please find below our point-by-point response to the points raised:

As you will see, ref #3 is satisfied with the revision while ref #2 raises a remaining concern that will have to be addressed either with the inclusion of additional data/analysis or with a moderation of the conclusion of the overlap between ESRP and QKI-5. I would therefore like to invite you to submit a final revision of the manuscript in which you address this issue as well as the following editorial points:

Referee #2:

“The authors have responded to most of my critiques, but the response to one of my major critiques is inadequately addressed. This relates to the degree of overlap or combinatorial regulation of EMT splicing switches by QKI and ESRPs. The response does not address several of the main points of the critique and they basically do not change their conclusion by merely adding "appear to be" largely distinct subsets of alternative splicing events. The authors do not list ADD3 and NUMB as ESRP regulated events, I believe, by misinterpreting the data in the Yang paper. These EMT associated events, while not passing the statistical thresholds as regulated events in ESRP depleted cells by RNA-Seq, nonetheless were confirmed to be regulated by them by RT-PCR. For ADD3, the changes in splicing was relatively small, but still statistically significant. Such examples illustrate the false negative issue for splicing analysis in RNA-Seq datasets that was pointed out in the critique. The authors also did not consider or compare to other ESRP depletion datasets or those of orthologous events from RNA-Seq analysis of mouse ESRP1/2 KO mice as suggested in the critique. Thus, of the 11 QKI validated events they validated, in fact 8 of the 11 have been shown to also be regulated by ESRPs/Esrps (ADD3, DEPDC1, EXOC1, NUMB, CLSTN1, MPRIP, MYO18A, and NYFA). The authors ought to point out all of these co-regulated events. Based on these validated examples the statement that their activities appear to be largely distinct is not supported. If the authors wish to retain this claim, they would need to carry out a larger analysis of regulated by both factors in the same cell system as suggested in the critique. Alternatively, they could modify the statement to simply state that the degree of overlap in co-regulated targets would need to be determined by global splicing analysis in the same cell type(s).

We thank the reviewer for this clarification, and agree that to definitely determine the degree of overlap between ESRP and QKI-5 regulated splice events we would need to carry out global splicing analysis in the same cell type. So as not to over-interpret comparisons between different experiments, we have elected to remove the statement “Comparison of our data with this earlier study carried out in a different cell line indicates that although QKI and ESRP1/2 co-regulate a subset of events in opposing directions, their activities appear to be largely distinct (Appendix Fig. S1B).” and the associated Appendix Fig. S1B.

-> We noticed that a call-out for fig 4G is missing.

This has now been included on page 8.

-> Please change the nomenclature for the tables, they should be Table EV1, Table EV2 etc. The Appendix nomenclature is only used for information that is presented in the appendix pdf. Please also update the table call-outs in the main manuscript file. See our online guide to authors for more detail <http://emboj.emboress.org/authorguide>

We have changed the nomenclature of the Tables as suggested.

-> Please update the reference style to fit with the journal guidelines (author1 et al in main manuscript, not author1, author2 et al in main manuscript).

We have changed the reference style to fit with the journal guidelines.

-> We generally encourage the publication of source data, particularly for electrophoretic gels and blots, with the aim of making primary data more accessible and transparent to the reader. We would need 1 file per figure (which can be a composite of source data from several panels) in jpg, gif or

PDF format, uploaded as "Source data files". The gels should be labelled with the appropriate figure/panel number, and should have molecular weight markers; further annotation would clearly be useful but is not essential. These files will be published online with the article as a supplementary "Source Data". Please let me know if you have any questions about this policy.

We have included source data for all of our blots.

-> Our production/data editors have gone through the manuscript file and noticed a couple of points in the figure legends that need clarification (see attached document). Please incorporate these changes in the word document with track changes activated. For the point about the asterisk is the miRNA names, I realise that this is a nomenclature issue but it may be beneficial to use the -5p, 3p names for these instead to avoid confusion.

We have addressed these "track changed" points in the figure legends. We defined all the box and whiskers plots more clearly (Figs 1D-F, 4B-C, and EV1C-D), described the miRNA * annotation (Fig 1H), defined the red box (Fig EV2C), and removed the error bars from the qPCR technical replicates in Fig EV3D.

Corresponding Author Name: Philip Gregory

Journal Submitted to: The EMBO Journal

Manuscript Number: EMBOJ-2018-99016